# Aberrant cortical activity, functional connectivity, and neural assembly architecture after photothrombotic stroke in mice

**Mischa Vance Bandet[1,2,3], Ian Robert Winship[1,2,3]***

[1]Neuroscience and Mental Health Institute, University of Alberta, Edmonton, Canada; [2]Neurochemical Research Unit, University of Alberta, Edmonton, Canada; [3]Department of Psychiatry, University of Alberta, Edmonton, Canada

**\*For correspondence:**
iwinship@ualberta.ca

**Competing interest:** The authors declare that no competing interests exist.

**Abstract** Despite substantial progress in mapping the trajectory of network plasticity resulting from focal ischemic stroke, the extent and nature of changes in neuronal excitability and activity within the peri-infarct cortex of mice remains poorly defined. Most of the available data have been acquired from anesthetized animals, acute tissue slices, or infer changes in excitability from immunoassays on extracted tissue, and thus may not reflect cortical activity dynamics in the intact cortex of an awake animal. Here, in vivo two-photon calcium imaging in awake, behaving mice was used to longitudinally track cortical activity, network functional connectivity, and neural assembly architecture for 2 months following photothrombotic stroke targeting the forelimb somatosensory cortex. Sensorimotor recovery was tracked over the weeks following stroke, allowing us to relate network changes to behavior. Our data revealed spatially restricted but long-lasting alterations in somatosensory neural network function and connectivity. Specifically, we demonstrate significant and long-lasting disruptions in neural assembly architecture concurrent with a deficit in functional connectivity between individual neurons. Reductions in neuronal spiking in peri-infarct cortex were transient but predictive of impairment in skilled locomotion measured in the tapered beam task. Notably, altered neural networks were highly localized, with assembly architecture and neural connectivity relatively unaltered a short distance from the peri-infarct cortex, even in regions within 'remapped' forelimb functional representations identified using mesoscale imaging with anaesthetized preparations 8 weeks after stroke. Thus, using longitudinal two-photon microscopy in awake animals, these data show a complex spatiotemporal relationship between peri-infarct neuronal network function and behavioral recovery. Moreover, the data highlight an apparent disconnect between dramatic functional remapping identified using strong sensory stimulation in anaesthetized mice compared to more subtle and spatially restricted changes in individual neuron and local network function in awake mice during stroke recovery.

## eLife assessment

This **important** study sheds light on several apparent discrepancies observed across animal studies examining neuroimaging biomarkers of functional recovery following focal ischemia. Using 2-photon imaging of calcium activity in awake mice, the authors show **compelling** evidence that deficits in neuronal activity and functional connectivity after photothrombosis occur within a very small distance from the infarct (<750 microns) whereas these measures were relatively unaltered more distally, even those typically implicated with functional remapping of the forelimb representation in anaesthetized animals. These findings reveal a complex spatiotemporal relationship between perilesional neuronal

network function and behavioral recovery that is more nuanced than previously reported, and motivates the need for better criteria for what is considered remapping.

## Introduction

Investigations of the excitability of neurons in the peri-infarct cortex during recovery from ischemic stroke have yielded a complex and at times contradictory data set. Peri-infarct hyper-excitability was suggested by data from anesthetized rats that exhibited elevated spontaneous multiunit firing within peri-infarct regions 3–7 days after stroke (*Schiene et al., 1996*). Similarly, downregulated GABAergic inhibition in peri-infarct cortex (*Schiene et al., 1996*; *Mittmann et al., 1998*; *Neumann-Haefelin et al., 1998*; *Neumann-Haefelin et al., 1999*; *Qü et al., 1998a*; *Redecker et al., 2002*), degeneration of parvalbumin-positive inhibitory interneurons (*Neumann-Haefelin et al., 1998*; *Luhmann et al., 1995*), decreased paired pulse inhibition (*Buchkremer-Ratzmann and Witte, 1997*; *Domann et al., 1993*; *Fujioka et al., 2004*), increased NMDA-receptor-mediated excitation (*Qü et al., 1998b*; *Que et al., 1999*), and a downregulation of KCC2 *Jaenisch et al., 2016*; *Jin et al., 2005*; *Khirug et al., 2021*; *Martín-Aragón Baudel et al., 2017*; *Schulte et al., 2018* have suggested that the peri-infarct cortex is hyper-excitable during recovery, contributing to increased risk of epileptogenesis in the post stroke brain (*Jaenisch et al., 2016*; *Witte and Freund, 1999*). Conversely, several groups have reported that the sensory-evoked responsiveness of the peri-infarct cortex is diminished during recovery from focal photothombotic stroke (*Brown et al., 2009*; *Chen et al., 2012*; *Lim et al., 2014*; *Sigler et al., 2009*; *Sweetnam and Brown, 2013*; *Sweetnam et al., 2012*; *Winship and Murphy, 2008*). Even after months of recovery following forelimb (FL) representation targeted stroke, the remapped representation of the FL has been shown to display prolonged modes of activation with lower amplitude (*Brown et al., 2009*) and reduced temporal fidelity (*Sweetnam and Brown, 2013*).

Almost all studies that have examined post-stroke cortical responsiveness have used surgical preparations with anesthetized animals. Anesthesia disrupts cortical activity dynamics and functional connectivity between cortical areas (*Cramer et al., 2019*; *Grandjean et al., 2014*; *Jonckers et al., 2014*; *Kalthoff et al., 2013*; *Nasrallah et al., 2014*), and has been shown to reduce the magnitude and spread of cortical activation from sensory stimuli (*Chapin and Lin, 1984*; *Chen et al., 2005*; *Devonshire et al., 2010*; *Lissek et al., 2016*; *Ebner and Kaas, 2015*; *Friedberg et al., 1999*; *Faggin et al., 1997*; *Moore and Nelson, 1998*; *Nicolelis and Chapin, 1994*). It further reduces the potential contribution of corollary discharge and reafference due to voluntary action in modulating sensory input and cortical responses (*Crapse and Sommer, 2008*). It has also recently been shown that the propagation pattern of cortical activity differs between evoked and spontaneous activity, with spontaneous activity showing more complex trajectories and lower activity amplitudes (*Afrashteh et al., 2021*). Somatosensory responses are known to be modulated by the relevance of stimuli to behavior and task performance (*Dionne et al., 2013*; *Johansen-Berg et al., 2000*; *Nelson et al., 2004*; *Staines et al., 2002b*; *Gomez-Ramirez et al., 2016*; *Scaglione et al., 2014*), and by corollary feedback during (*Yu et al., 2016*), or even before (*London and Miller, 2013*; *Nelson, 1987*; *Bensmaia and Helms Tillery, 2014*), the onset of movement. Together, these studies point to a complex, interconnected system that modulates the activity of somatosensory networks in the awake behaving animal that is either not present, or is altered, in studies of evoked cortical activity under the anesthetized state. While monitoring cortical activity in awake mobile animals presents methodological complexity, it avoids confounds associated with the anesthetized state and provides unique insight into evolving patterns of network activity that may be unique to the awake state.

To investigate neural activity patterns during stroke recovery while avoiding the potential confounds of anesthesia, the present work used two photon $Ca^{2+}$ imaging within and distal to the peri-infarct region of a focal photothrombotic stroke lesioning the FL primary somatosensory cortex in mice. We used awake, behaving but head-fixed mice in a mobile homecage to longitudinally measure cortical activity, then used computational methods to assess functional connectivity and neural assembly architecture at baseline and each week for 2 months following stroke. Behavioral recovery from stroke was measured in the same animals on a tapered beam task and string pull task to determine the time course of sensorimotor deficits related to changes in cortical activity and network architecture. Our goal was to use longitudinal imaging and advanced computational analyses to define network changes in somatosensory cortex associated with sensorimotor recovery. We show that despite disturbances

to the widefield topology and amplitude of the sensory-evoked FL map (acquired under anesthetic), it is only within the immediate peri-infarct region after FL targeted stroke that neural activity, functional connectivity, and neural assembly architecture are disrupted. We demonstrate that the peri-infarct somatosensory cortex displays a marked reduction in neural assembly number, an increase in neural assembly membership, and persistent alterations in assembly-assembly correlations, and that this occurs alongside transient deficits in both functional connectivity and neural activity. Notably, we show that reduced neuronal spiking is strongly correlated with a deficit in motor performance on the tapered beam task. Surprisingly, we also show that significant alterations in neuronal activity (firing rate), functional connectivity, and neural assembly architecture are absent within more distal regions of cortex as little as 750 µm from the stroke border, even in areas identified by regional functional imaging (under anaesthesia) as 'remapped' locations of sensory-evoked FL activity 8 weeks post-stroke.

## Results

The experimental timeline is illustrated in *Figure 1A*. Adult Thy1-GCaMP6S mice were implanted with chronic cranial windows and habituated on the floating homecage, tapered beam and string pull task (*Figure 1A–C*). The cFL and cHL somatosensory areas were mapped on the cortex using widefield Ca$^{2+}$ imaging of sensory-evoked activity in isoflurane anaesthetized mice (*Figures 1D and 2A*, see methods). Regions of interest for longitudinal awake two-photon Ca$^{2+}$ imaging of cellular activity (*Figure 1E*) were chosen based on these pre-stroke widefield Ca$^{2+}$ limb maps (*Figure 2*). The first imaging region of interest was located at the boundary of the pre-stroke cFL and cHL somatosensory maps (termed 'peri-infarct' region based on its proximal location to the photothrombotic infarct after stroke) and the second region as lateral to the pre-stroke cHL map (termed 'distal' region due to its distance from the stroke boundary). These regions were imaged at baseline, and each week following photothrombotic stroke. Notably, these regions of interest incorporated the predicted areas of remapping for the limb associated somatosensory representations after focal cFL cortex stroke based on previous studies (*Winship and Murphy, 2008*; *Winship and Murphy, 2009*). Photothrombosis was directed to the cFL somatosensory representation as identified by regional imaging. Resulting infarcts lesioned this region, and borders were delineated by a region of decreased light absorption 1 week post-stroke (*Figure 1D*, Top). Within the peri-infarct imaging region, cellular dysmorphia and swelling was visually apparent in some cells during two-photon imaging 1 week after stroke, but recovered over the 2-month post-stroke imaging timeframe (data not shown). These gross morphological changes were not visually apparent in the more distal imaging region lateral to the cHL.

### Altered sensory-evoked widefield Ca$^{2+}$ maps after stroke

Regional remapping of the limb-associated somatosensory cortex occurs over the weeks following focal cortical stroke (*Winship and Murphy, 2008*; *Winship and Murphy, 2009*). Here, widefield Ca$^{2+}$ imaging of the cranial window in isoflurane anaesthetized mice was performed during piezoelectric stimulation of the contralateral limbs prior to stroke (to define the region targeted for infarction as described above), and at the 8-week post-stroke timepoint (to define regional remapping of the forelimb representation; see *Figure 2A* and Methods). *Figure 2A* shows representative montages from a stroke animal illustrating the cortical cFL and cHL Ca$^{2+}$ responses to 1 s, 100 Hz limb stimulation of the contralateral limbs at the pre-stroke and 8-week post-stroke timepoints. The location and magnitude of the cortical responses changes drastically between timepoints, with substantial loss of supra-threshold activity within the pre-stroke cFL representation located anterior to the cHL map, and an apparent shift of the remapped representation into regions lateral to the cHL representation at 8 weeks post-stroke. A significant decrease in the cFL evoked Ca$^{2+}$ response amplitude was observed in the stroke group at 8 weeks post-stroke relative to pre-stroke (*Figure 2B*). This is in agreement with past studies (*Brown et al., 2009*; *Chen et al., 2012*; *Lim et al., 2014*; *Sigler et al., 2009*; *Sweetnam and Brown, 2013*; *Sweetnam et al., 2012*; *Winship and Murphy, 2008*), and suggests that cFL targeted stroke reduces forelimb evoked activity across the cFL somatosensory cortex in anaesthetized animals even after 2 months of recovery. There was no statistical change in the average size of cFL evoked representation 8 weeks after stroke (*Figure 2C*), but a significant posterior shift of the supra-threshold cFL map was detected (*Figure 2D*). Unmasking of previously sub-threshold cFL responsive cortex in areas posterior to the original cFL map at 8 weeks post-stroke could contribute

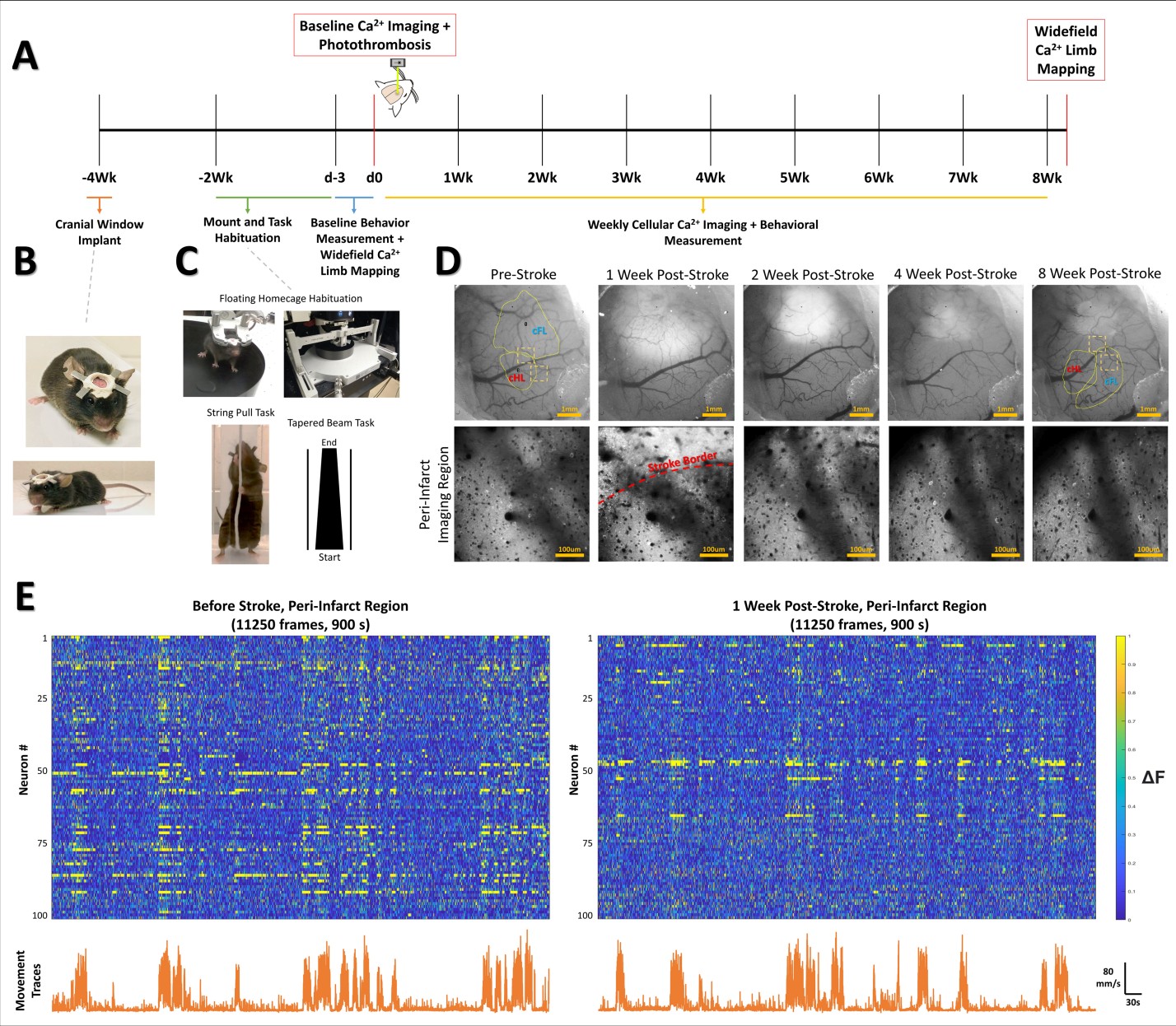

**Figure 1.** Experimental timeline and methods. (**A**) Timeline of experimental procedures, imaging times, and behavioral tests. (**B**) Glass cranial windows and headplates were implanted 2 weeks prior to beginning habituation protocols in Thy1-GCaMP6S mice. (**C**) Mice were habituated on the floating homecage, string pull and tapered beam for a period of 2 weeks prior to baseline measurements for behavior. Widefield Ca2 +response maps were used to determine locations for the imaging site located between the cFL and cHL sensory maps at baseline ('peri-infarct' imaging region) and imaging site lateral to the cHL at baseline ('distal' imaging region). For each $Ca^{2+}$ imaging session, both imaging regions were imaged for a period of 15 min while simultaneously tracking animal movement within the mobile homecage. After baseline $Ca^{2+}$ imaging, the somatosensory cFL map was targeted with photothrombosis. Behavioral testing on the string pull task and tapered beam task were performed weekly 1 day prior to the $Ca^{2+}$ imaging session. After the final cellular $Ca^{2+}$ imaging session at the 8-week time, stimulus-evoked widefield $Ca^{2+}$ imaging was once again performed to determine the cFL and cHL somatosensory maps. (**D**) Representative widefield and two photon calcium imaging at each weekly time. The area of decreased light absorption due to stroke damage is apparent, with its surface area decreasing over time. (**E**) Neuron $Ca^{2+}$ traces ($\Delta F/F_o$) for 100 neurons from one example animal pre-stroke (left panel) and 1 week post-stroke (right panel) in the peri-infarct somatosensory imaging region. Lower line graphs demonstrate the tracked movement (mm/s) within the floating homecage corresponding to the $Ca^{2+}$ traces above.

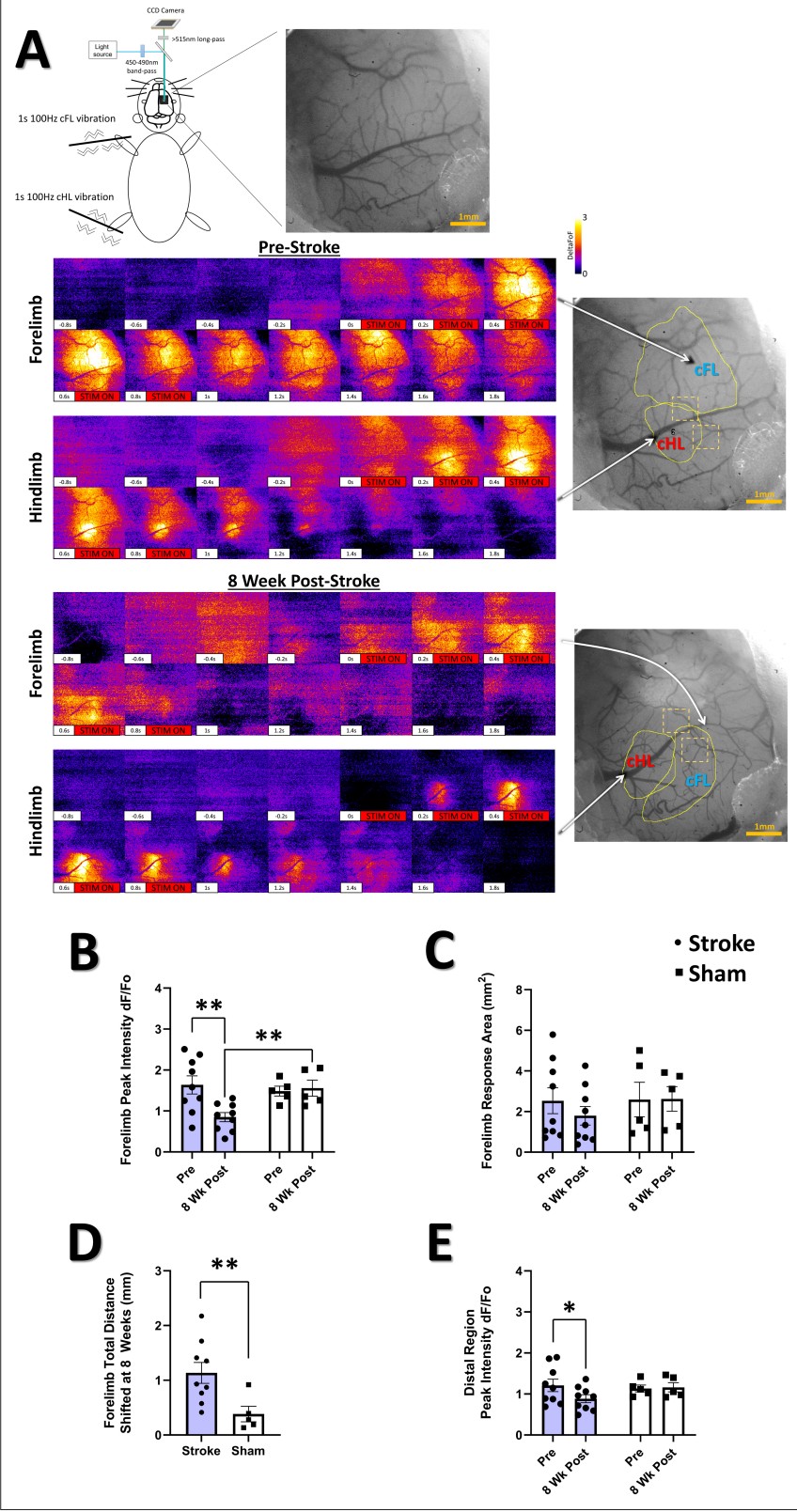

**Figure 2.** Photothrombotic stroke results in forelimb representation shift onto adjacent areas of cortex and altered sensory-evoked widefield Ca²⁺ response properties at 8 weeks post-stroke. (**A**) Pseudocolored (ΔF/F$_o$) montages of representative cFL and cHL responses in S1 of an anesthetized animal pre-stroke and 8-week post-stroke resulting from oscillatory stimulation (1 s, 100 Hz) of the cFL and cHL, respectively. At the 8-week post-stroke

*Figure 2 continued on next page*

*Figure 2 continued*

time, the cFL map has shifted posterior to its pre-stroke location into the area lateral to the cHL. (**B**) Mean peak forelimb $Ca^{2+}$ transient response intensity ($\Delta F/F_o$) measured from the thresholded cortical map area of the cFL. A significant interaction between group and time was found, with *post-hoc* tests showing a significant decrease in the peak cFL response at 8-week post-stroke compared to pre-stroke (p=0.0055) and compared to 8-week sham (p=0.0277). Mixed Effects Model, Time $F_{(1, 12)}$=4.151, p=0.0643; Group $F_{(1, 12)}$=1.909, p=0.1922; Interaction $F_{(1, 12)}$=6.010, p=0.0305. (**C**) Thresholded cFL response area ($mm^2$). No main effects or interaction was observed. (**D**) The stroke group was found to have significantly greater total distance shifted for the forelimb map at the 8-week time compared to sham (p=0.0080). (**E**) Mean peak forelimb $Ca^{2+}$ transient response intensity ($\Delta F/F_o$) measured from the distal region ROI lateral to the hindlimb response map. Post-hoc tests show a decrease in distal region peak response intensity at 8 weeks in the stroke group (p=0.0442). Mixed Effects Model, Time $F_{(1, 12)}$=2.054, p=0.1773; Group $F_{(1, 12)}$=0.4256, p=0.5264; Interaction $F_{(1, 12)}$=2.889, p=0.1150. Stroke N=9, Sham N=5. *p<0.05; **p<0.01; ***p<0.001.

---

to this apparent remapping. However, the amplitude of the cFL evoked widefield $Ca^{2+}$ response in this distal region at 8 weeks post-stroke remains reduced relative to pre-stroke activation (*Figure 2E*). Previous studies suggest strong inhibition of cFL evoked activity during the first weeks after photo-thrombosis (*Winship and Murphy, 2008*). Without longitudinal measurement in this study to quantify this reduced activation prior to 8 weeks post-stroke, we cannot differentiate potential remapping due to unmasking of the cFL representation that enhances the cFL-evoked widefield $Ca^{2+}$ response from apparent remapping that simply reflects changes in the signal-to-noise ratio used to define the functional representations. There were no group differences between stroke and sham groups in cHL evoked intensity, area, or map position (data not shown).

## Impaired performance on tapered beam but not string pull after focal somatosensory stroke

Post-stroke sensorimotor impairment was measured on the tapered beam task and string pull task. In the tapered beam, errors (slips off the beam), indicative of post-stroke impairment, were tracked as the number of left (affected) side slips, right (non-affected) side slips, and the distance to first slip (*Figure 3A*). We observed a significant interaction of group and time in the number of left side (contralesional) slips (*Figure 3B*), with post hoc comparisons confirming increased left slips at 1 week post-stroke compared to baseline and compared to 1 week in sham. Consistent with the unilateral nature of the photothrombotic damage to the cortex in this study, no main effects or interaction was seen in the number of right side slips (*Figure 3C*). No main effects or interaction was observed in the distance to first slip (*Figure 3D*). These results are consistent with a transient deficit in motor

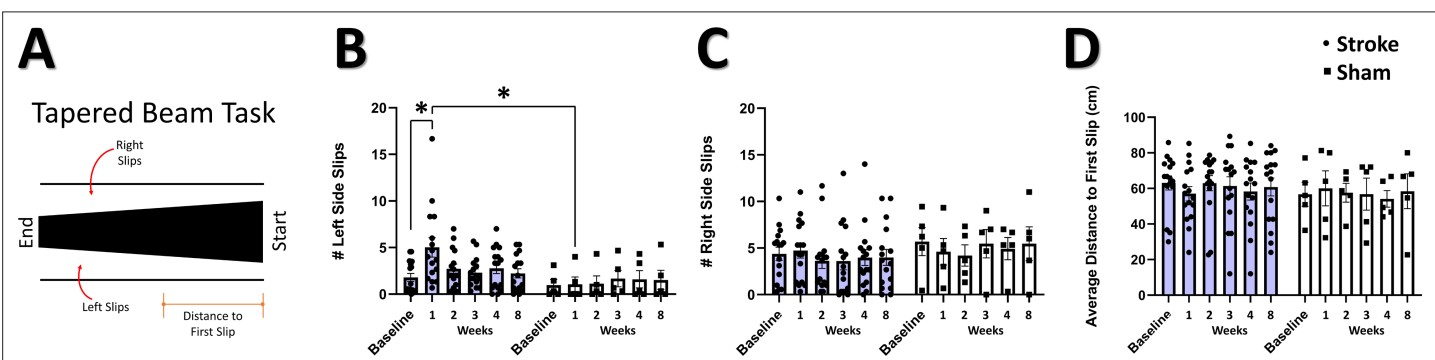

**Figure 3.** Impaired performance on tapered beam task after stroke. (**A**) Illustration of the tapered beam test with the three elements measured; left (contralesional) side slips, right side slips, and distance to first slip. (**B**) Mean number of left side slips. A significant interaction between time and group was observed, with *post-hoc* tests showing a greater number of left side slips at 1 week in the stroke group relative to the sham group (=0.0435), and in the stroke group at 1 week relative to baseline (p=0.0125). Mixed Effects Model, Time $F_{(2.007, 38.13)}$=2.178, p=0.1270; Group $F_{(1, 19)}$=2.260; p=0.1492; Interaction $F_{(5, 95)}$=2.914, p=0.0172. (**C**) Mean number of right-side slips. No main effects or interactions were observed. (**D**) Distance to first slip. No main effects or interactions were observed. Stroke N=16, Sham N=5. *p<0.05; **p<0.01; ***p<0.001.

The online version of this article includes the following figure supplement(s) for figure 3:

**Figure supplement 1.** String pull task does not detect a behavioral deficit after stroke.

behavior on the tapered beam during the first week after stroke induced by focal cortical lesion to the sensorimotor cortex (*Ardesch et al., 2017*; *Zhao et al., 2005*), as opposed to more sustained deficits observed in models of middle cerebral artery occlusion (*Lipsanen et al., 2011*; *Schallert et al., 2002*). In contrast to previous studies investigating string pull after stroke (*Blackwell et al., 2018c*), we did not detect a significant effect of photothrombosis on any kinematic parameters during string pulling (*Figure 3—figure supplement 1*).

## Firing rate of neurons in the peri-infarct cortex correlates with performance on the tapered beam task

The firing rate of cortical somatosensory neurons in our awake behaving animals (*Figure 4A*) was calculated using a custom Matlab algorithm to identify 'significant' calcium transients (*Romano et al., 2015*; *Romano et al., 2017*) and the mean firing rate of the sampled neural population during periods of movement and periods of rest in the mobile homecage. The average firing rate in the peri-infarct region during recovery in movement and rest is illustrated in *Figure 4B and C*, respectively. A significant main effect of group on firing rate was detected at rest (*Figure 4C*). Post hoc comparisons confirmed significantly reduced average firing rate when moving and at rest at 1 week in the stroke group compared to pre-stroke (*Figure 4B and C*, respectively). Notably, firing rate at 1 week post-stroke during movement and at rest were significantly negatively correlated with the number of left slips in the tapered beam at the same 1 week timepoint (*Figure 4D and E*). No main effects or significant interaction was observed in the distal imaging region for either movement or at rest (*Figure 4F and G*, respectively). Firing rate during movement and at rest at 1 week within the distal region was also not significantly correlated with left slips on the tapered beam at the same timepoint (*Figure 4H and I*, respectively). These results indicate decreased neuronal spiking 1 week after stroke in regions immediately adjacent to the infarct, but not in distal regions, that is strongly related to sensorimotor impairment. This finding runs contrary to a previous report of increased spontaneous multi-unit activity as early as 3–7 days after focal photothrombotic stroke in the peri-infarct cortex (*Schiene et al., 1996*), but is in agreement with recent single-cell calcium imaging data demonstrating reduced sensory-evoked activity in neurons within the peri-infarct cortex after stroke (*Zeiger et al., 2021*; *Motaharinia et al., 2021*). While the majority of neuronal activity occurred during movement, correlations between firing rate and animal movement were low and did not vary between stroke and sham groups (*Figure 4—figure supplement 1*).

## Altered functional connectivity in the peri-infarct somatosensory cortex

Studies in humans and animals using mesoscale imaging methods indicate that stroke disrupts functional connectivity between widespread cortical regions (*Cramer et al., 2019*; *Bauer et al., 2014*; *Blaschke et al., 2021*; *Carter et al., 2010*; *Carter et al., 2012*; *Hakon et al., 2018*; *Longo et al., 2022*; *Olafson et al., 2021*; *Rehme and Grefkes, 2013*; *Siegel et al., 2016*; *Silasi and Murphy, 2014*; *Urbin et al., 2014*). However, few studies have examined functional connectivity at the level of local neural populations after focal stroke (*Latifi et al., 2020*; *Bechay et al., 2022*), and none have examined the primary somatosensory cortex. To define functional connectivity in local neural networks within the peri-infarct somatosensory cortex, we plotted the functional connectivity of the neural population in terms of the strength of their cell-cell correlations (*Figure 5*) and quantified the properties of these connections (*Figure 6*). We observed a significant loss of functional connectivity 1 week after stroke within the peri-infarct region. This was visually apparent in the stroke group functional connectivity plots at the 1 week timepoint (*Figure 5A and B*) compared to the same location in sham animals (*Figure 5E and F*). Notably, no profound loss of functional connectivity is visually apparent in the distal region in the stroke (*Figure 5C and D*) or sham (*Figure 5G and H*) groups. Within the peri-infarct region, we observed a statistical trend of a main effect of time in the average number of 'significant' functional connections per neuron (*Figure 6A*; see Materials and methods). Post hoc comparisons confirmed a significantly reduced average number of significant connections per neuron at 1 week post-stroke compared to pre-stroke and to 1 week sham. We observed a significant main effect of time on the average number of strong connections ($r>.3$) per neuron (*Figure 6B*). Post hoc comparisons confirmed strong connections were significantly reduced 1 week post-stroke relative to pre-stroke. In the distal region, no main effects or interaction was identified in the average number of significant connections per neuron (*Figure 6D*), highlighting regionally specific connectivity

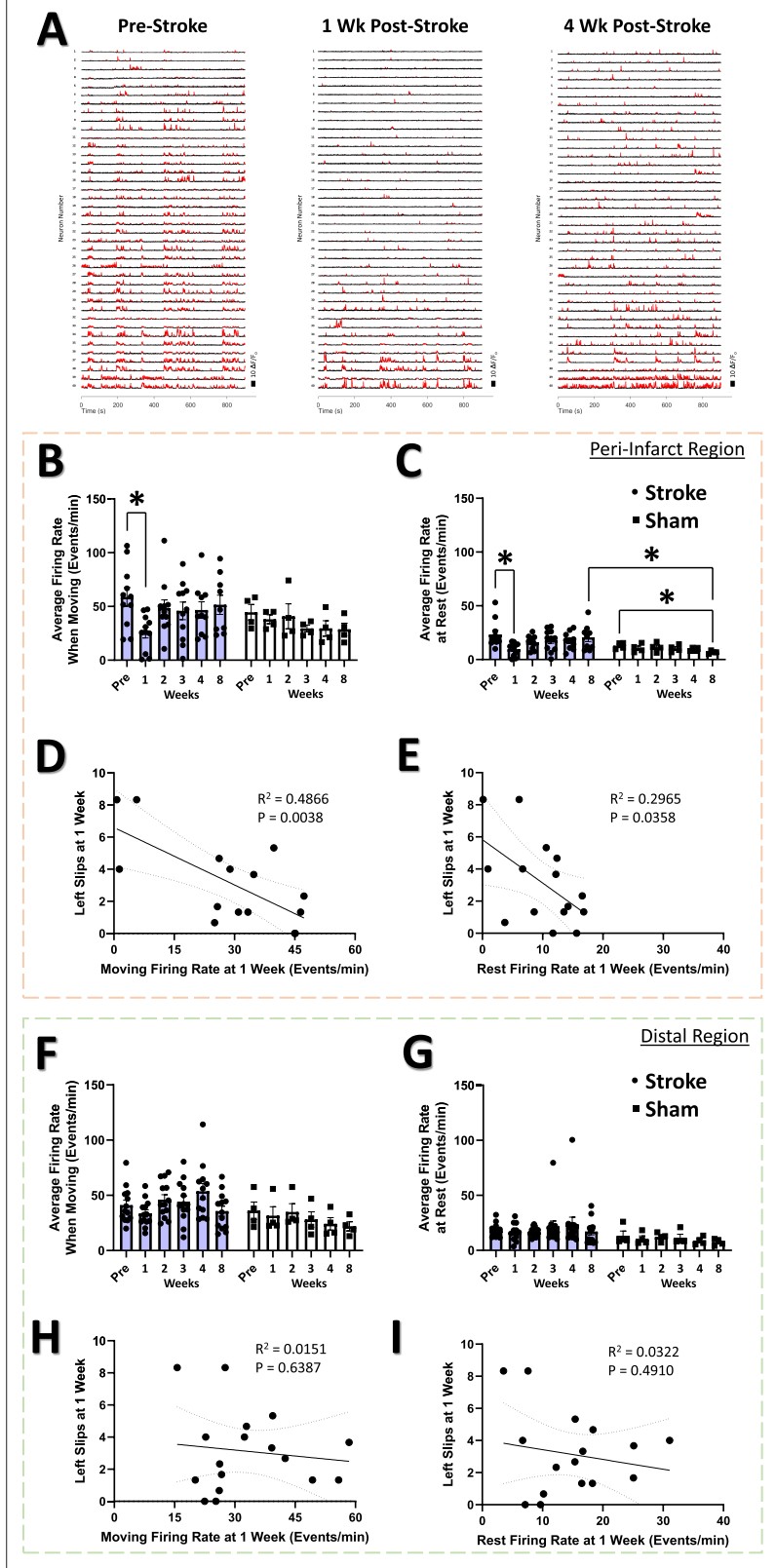

**Figure 4.** Firing rate of neurons in the peri-infarct cortex correlates with performance on the tapered beam task. (**A**) Example Ca$^{2+}$ traces for 40 example neurons at each time point from the peri-infarct imaging region for one example animal in the stroke group. Significant Ca$^{2+}$ transients are highlighted by red segments. A decrease in the number of Ca$^{2+}$ transients is visually apparent at 1 week post-stroke. (**B**) In the peri-infarct imaging region, no

*Figure 4 continued on next page*

Figure 4 continued

main effects or interaction was seen in the average firing rate during animal movement, however *post-hoc* analysis indicates significantly decreased average firing rate at 1 week in the stroke group relative to pre-stroke (p=0.0124). Mixed Effects Model, Time $F_{(3.217, 39.89)}$=1.629, p=0.1953; Group $F_{(1, 13)}$=1.106, p=0.3121; Interaction $F_{(5, 62)}$=1.079, p=0.3810. A significant main effect of group was seen in the average firing rate at rest (**C**), with post hoc tests showing a significant decrease a 1 week post-stroke compared to pre-stroke (p=0.0325), a significant decrease at 8 weeks post-sham compared with pre-sham (p=0.0280), and significantly lower at 8 weeks post-sham compared with 8 weeks post-stroke (p=0.0439). Mixed Effects Model, Time $F_{(2.297, 28.48)}$=1.360, p=0.2742; Group $F_{(1, 13)}$=6.058, p=0.0286; Interaction $F_{(5, 62)}$=1.169, p=0.3346. Average firing rate 1 week post-stroke was negatively correlated with the number of left slips in the tapered beam at the same 1 week time during both movement ($R^2$=0.4866, p=0.0038) (**D**) and rest ($R^2$=0.2965, p=0.0358) (**E**). In the distal imaging region, no main effects or interaction were found in the average firing rate for movement (**F**) or at rest (**G**). No correlation was observed between the moving firing rate at 1 week in the distal region and the number of left slips on the tapered beam ($R^2$=0.0151, p=0.6387) (**H**). No correlation was observed between the resting firing rate at 1 week in the distal region and the number of left slips on the tapered beam ($R^2$=0.0322, p=0.4910) (**I**). Peri-infarct region Stroke N=11, Sham N=4. Distal region Stroke N=13, Sham N=4. *p<0.05; **p<0.01; ***p<0.001.

The online version of this article includes the following figure supplement(s) for figure 4:

**Figure supplement 1.** Average neuron correlation with animal movement is not detectably altered by stroke.

changes in individual neurons after stroke. However, we observed a significant main effect of time on the average number of strong ($r$>.3) connections per neuron in the distal region (*Figure 6E*), with *post-hoc* comparisons confirming that strong connections were significantly reduced 8 weeks post-stroke relative to pre-stroke. The deficit in significant connections between neurons for the peri-infarct region is consistent with a persistent deficit in functional connectivity lasting up to 30 days that was identified within the primary and supplementary motor cortex of mice using in vivo imaging after focal motor cortex stroke (*Latifi et al., 2020*; *Bechay et al., 2022*). However, network density, defined as the ratio of measured significant functional network connections to the number of total possible network connections if all neurons were interconnected, was not decreased in either peri-infarct or distal regions in our study (*Figure 6C and F*, respectively). This is in contrast with previous research in the primary and supplementary motor cortex after focal motor cortex stroke that has demonstrated deficits in network density lasting up to 3 weeks post-stroke (*Bechay et al., 2022*).

## Neural assembly structure is disrupted in the peri-infarct region after stroke

The neural assembly hypothesis defines coordinated activity in groups of neurons as the basis for the representation of external and internal stimuli within the brain (*Buzsáki, 2010*; *Buzsáki and Watson, 2012*; *Palm et al., 2014*). Likewise, perturbations within the structure and function of assemblies may signify aberrant processing of stimuli within neural networks, and is believed to play a significant role in the dysfunction observed in brain damage and neuropsychiatric diseases (*Buzsáki and Watson, 2012*). Although the use of sophisticated methods for detecting neural assemblies in large populations of neurons has gained increasing popularity with the advent of large population recordings using $Ca^{2+}$ imaging (*Mölter et al., 2018*), no previous studies have determined the longitudinal effects of stroke on the architecture of neural assemblies in cortex. To determine if focal stroke to the cFL cortex affects the properties of neural assemblies, we used a PCA-Promax procedure (*Romano et al., 2015*; *Romano et al., 2017*; *Mölter et al., 2018*) for neural assembly detection to determine assembly size, population membership, assembly quantity, and overlap in assembly membership. The PCA-Promax method for determining neural assemblies allows neurons to participate in more than a single assembly (*Romano et al., 2015*; *Romano et al., 2017*), a property necessary to examine the ability of neural assemblies to dynamically form with varying members of the population for only brief periods of time (*Romano et al., 2015*; *Romano et al., 2017*; *Mölter et al., 2018*). *Figure 7* shows representative neuronal $Ca^{2+}$ fluorescence images with neural assemblies color coded and overlaid for each timepoint for the peri-infarct imaging region and for the distal imaging region for an example stroke and sham animal (*Figure 7A and B*, respectively). We observed a significant main effect of group on the number of assemblies in the peri-infarct imaging region (*Figure 8A*). Post hoc comparisons confirmed significantly reduced number of assemblies 1- and 4 weeks after stroke relative to

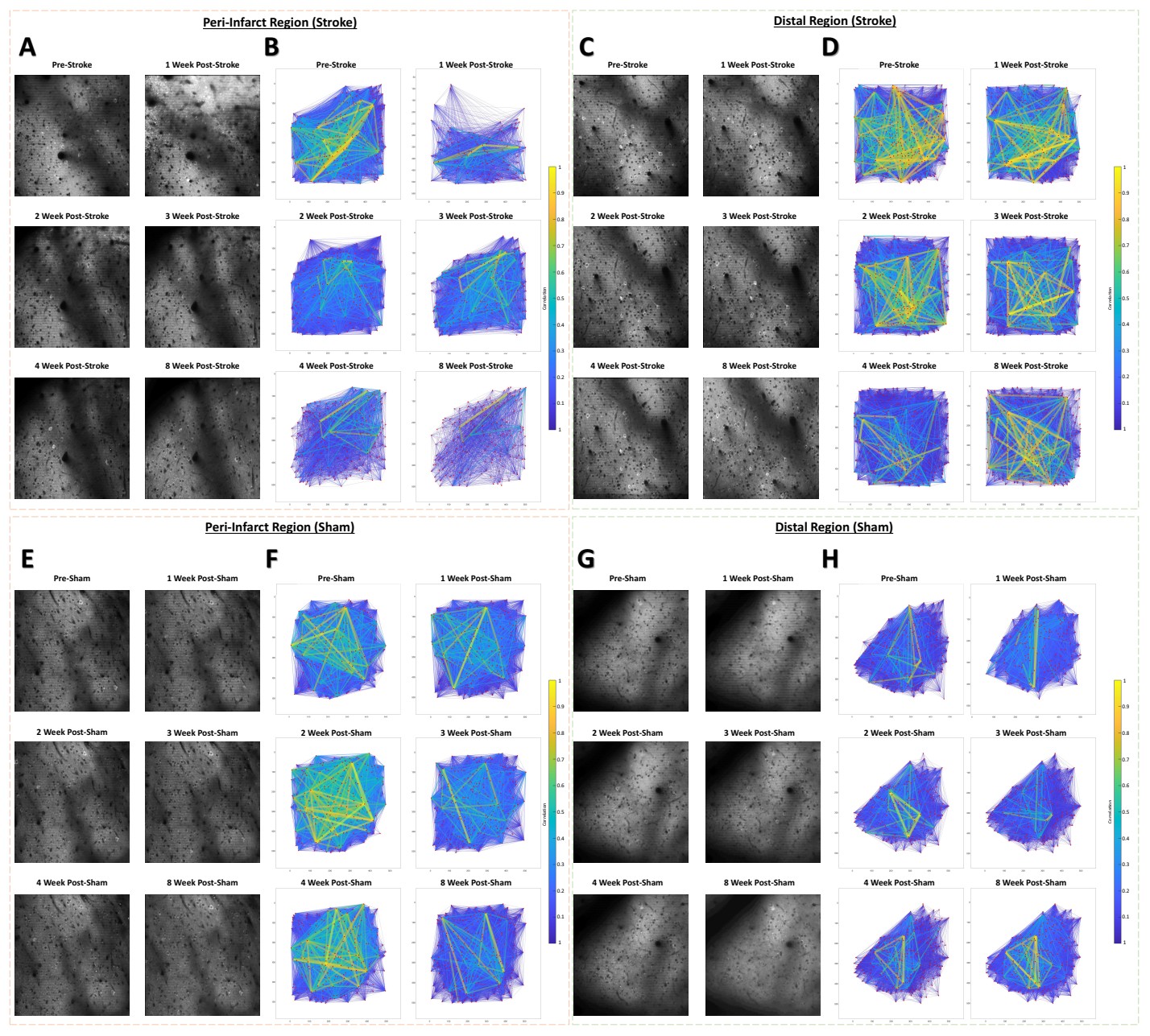

**Figure 5.** Example cellular GCaMP6S Ca²⁺ fluorescence images and population functional connectivity plots. (**A–D**) Ca²⁺ fluorescence images and functional connectivity plots from the peri-infarct imaging region and the distal imaging region for one example stroke group animal. Population functional connectivity plots demonstrate neurons as red dots, and the functional connection strength as lines with color and line width determined by the strength of the correlation between neurons. Notable cortical damage is visible in the upper portion of the 1 week post-stroke Ca²⁺ fluorescence image in the peri-infarct imaging region (**A**), which notably improves over the 2–4 week times. A notable loss of functional connections and loss in the number of connections with strong correlation (>0.3) is visible in the functional connectivity plots at 1 week post-stroke in the peri-infarct imaging region (**B**). In the distal imaging region post-stroke, little structural change is visible in the cellular Ca²⁺ fluorescence images (**C**), and less change to functional connectivity of the population over time (**D**). (**E–H**) Ca²⁺ fluorescence images and functional connectivity plots from the peri-infarct imaging region and the distal imaging region for one example sham group animal. Little structural changes within the cellular Ca²⁺ fluorescence images and in the functional connectivity plots are visually apparent relative to the example stroke animal.

sham controls (*Figure 8A*). Post hoc comparisons also indicated a significant increase in the average number of neurons per assembly at 1- and 2 weeks post-stroke relative to pre-stroke (*Figure 8B*). We further observed a statistical trend of a main effect of group in the average percent of the population per assembly (*Figure 8C*), with post hoc tests indicating a statistical trend for 1 week post-stroke

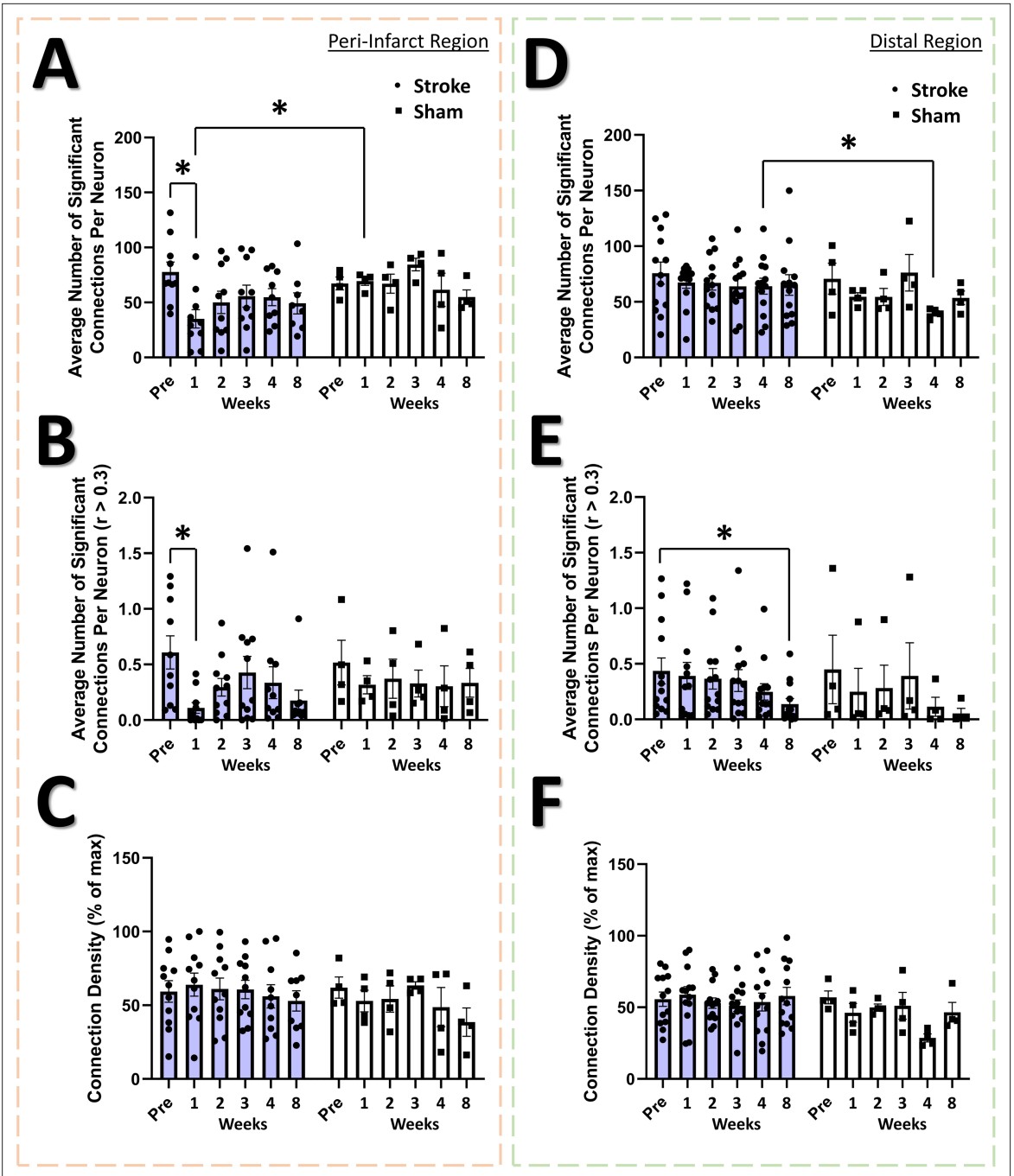

**Figure 6.** Altered functional connectivity in the peri-infarct somatosensory cortex. In the peri-infarct imaging region, a trend of a main effect of time was found in the average number of significant connections per neuron (**A**), with post hoc tests showing a significant decrease at 1 week post-stroke relative to pre-stroke (p=0.0373) and relative to 1 week post-sham (=0.0178). Mixed Effects Model, Time $F_{(3.559, 40.58)}$=2.213, p=0.0914; Group $F_{(1, 12)}$=1.716, p=0.2147; Interaction $F_{(5, 57)}$=1.820, p=0.1233. A significant main effect of time was found in the average number of significant connections per neurons with correlation greater than 0.3 (**B**), with post hoc tests indicating a significant decrease at 1 week post-stroke compared to pre-stroke (p=0.0398). Mixed Effects Model, Time $F_{(3.463, 41.55)}$=2.735, p=0.0484; Group $F_{(1, 13)}$=0.0964, p=0.7610; Interaction $F_{(5, 60)}$=0.8633, p=0.5111. No significant main effects or interaction is seen in the connection density for the peri-infarct imaging region (**C**). In the distal imaging region, no significant main effects or interaction is seen in the average number of significant connections per neuron (**D**), however post hoc tests indicate significantly lower connections per neuron at 4 weeks in the sham group relative to 4 weeks post-stroke (p=0.0360). Mixed Effects Model, Time $F_{(2.991, 44.87)}$=1.314, p=0.2815; Group $F_{(1, 15)}$=1.138, p=0.3030; Interaction $F_{(5, 75)}$=0.8022, p=0.5516. A significant main effect of time was observed for the average number of significant connections with correlation greater than 0.3 in the distal region (**E**), with post hoc tests indicating a significant decrease at 8 weeks post-stroke relative to pre-stroke (p=0.0189) Mixed Effects Model, Time $F_{(2.909, 43.06)}$=4.451, p=0.0088; Group $F_{(1, 15)}$=0.1272, p=0.7263; Interaction $F_{(5, 74)}$=0.8692, p=0.3676. No

*Figure 6 continued on next page*

*Figure 6 continued*

significant main effects or interaction is seen in the connection density (**F**). Peri-infarct region Stroke N=11, Sham N=4. Distal region Stroke N=13, Sham N=4. *p<0.05; **p<0.01; ***p<0.001.

compared with pre-stroke. These data suggest that after stroke, neurons in peri-infarct cortex are functionally grouped into fewer distinct assemblies, with more neurons per assembly and more of the local network in the same assembly. For the distal imaging region, no differences in neural assembly properties were observed within and between the stroke and sham groups (*Figure 8D–E*).

## Correlation between neural assembly activations is increased after stroke

If differential activation of neural assemblies in the naïve brain relates to representation of different aspects of the external world (*Romano et al., 2015*; *Buzsáki, 2010*; *Palm et al., 2014*; *Deolindo et al., 2017*) then changes in the relationship between the activation patterns of assemblies may predict disrupted processing of sensory information within neural networks and how these networks differentially represent external stimuli. An increase in assembly size and reduction in assembly number might reflect a mechanism to replace lost function and increase somatic sensitivity, but may come at the cost of reduced stimulus specificity/fidelity within the network. The activation magnitude of assemblies can be correlated between assemblies over the recording period (*Figure 9A*). A significant main effect of group was detected in the assembly-assembly correlation coefficient (*Figure 9B*) in the peri-infarct

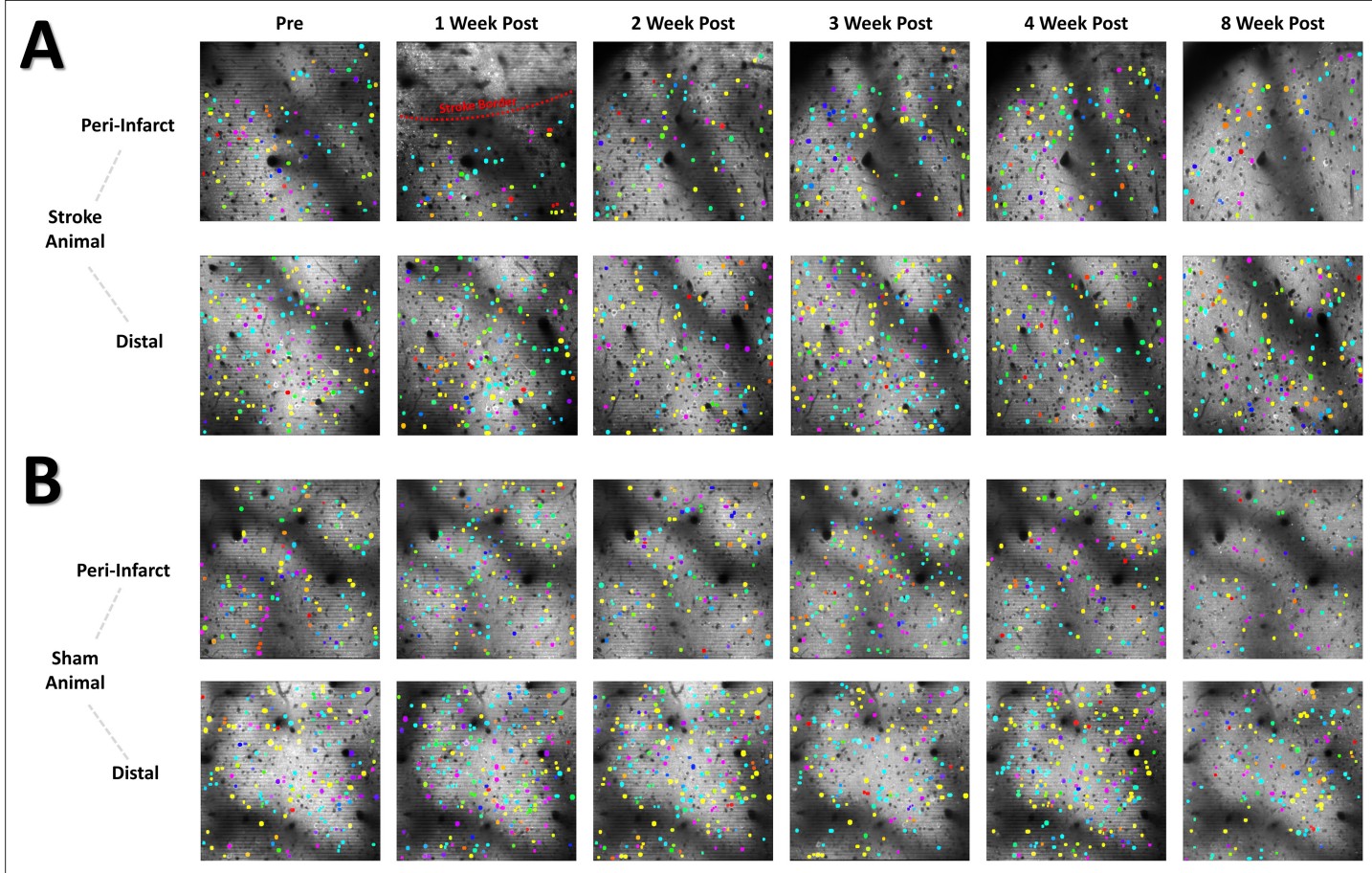

**Figure 7.** Color coded neural assembly plots depict altered neural assembly architecture after stroke in the peri-infarct region. (**A**) Representative cellular Ca²⁺ fluorescence images with neural assemblies color coded and overlaid for each timepoint. Neurons belonging to the same assembly have been pseudocolored with identical color. A loss in the number of neural assemblies after stroke in the peri-infarct region is visually apparent, along with a concurrent increase in the number of neurons for each remaining assembly. (**B**) Representative sham animal displays no visible change in the number of assemblies or number of neurons per assembly.

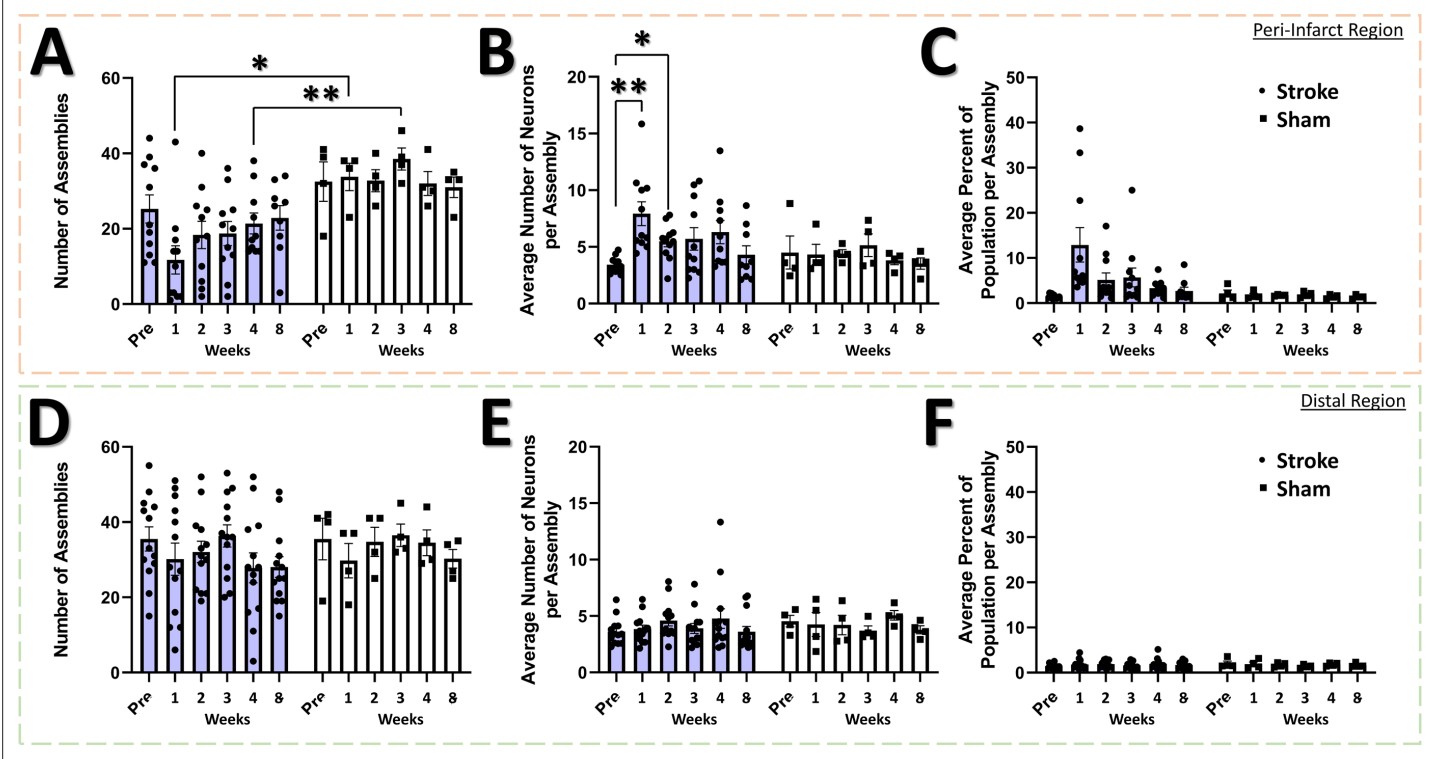

**Figure 8.** Neural assembly structure is disrupted in the peri-infarct region after stroke. In the peri-infarct imaging region, a significant main effect of group was found for the number of assemblies (**A**), with *post-hoc* tests showing a significant decrease in the number of assemblies at the 1 week and 3 week times for the stroke group compared to sham (p=0.0114 and 0.0058, respectively). Mixed Effects Model, Time $F_{(2.883, 35.75)}$=0.8231, p=0.4857; Group $F_{(1, 13)}$=11.90, p=0.0043; Interaction $F_{(5, 62)}$=1.496, p=0.2040. No main effects or interaction was observed in the average number of neurons per assembly (**B**), however post hoc tests indicated significantly higher number of neurons at 1- and 2 weeks post-stroke compared to pre-stroke (p=0.0099 and 0.0190, respectively). Mixed Effects Model, Time $F_{(3.405, 42.22)}$=1.865, p=0.1435; Group $F_{(1, 13)}$=2.018, p=0.1790; Interaction $F_{(5, 62)}$=1.728, p=0.1415. A trend of a main effect of group was found in the average percent of the population per assembly (**C**), with post hoc tests showing a trend at 1 week post-stroke compared to pre-stroke (p=0.0683) and compared to 1 week post-sham (p=0.0863). Mixed Effects Model, Time $F_{(1.711, 21.22)}$=1.738, p=0.2023; Group $F_{(1, 13)}$=3.807, p=0.0729; Interaction $F_{(5, 62)}$=1.765, p=0.1333. No main effects or interaction was observed in the distal imaging region for number of assemblies (**D**), average number of neurons per assembly (**E**), or average percent of the population per assembly (**F**). Peri-infarct region Stroke N=11, Sham N=4. Distal region Stroke N=13, Sham N=4. *p<0.05; **p<0.01; ***p<0.001.

cortex, with *post-hoc* tests indicating an increase in the correlation at 1 week post-stroke relative to pre-stroke and relative to 1 week in sham animals. No main effects or interaction was observed for the assembly-assembly correlation coefficient in the distal region (*Figure 9C*), suggesting that changes to assembly-assembly correlation were confined to the peri-infarct cortex. This increased tendency for functionally connected assemblies to be co-active after stroke may reduce the signal to noise ratio for stimulus representation within the network and may reduce stimulus specificity within peri-infarct cortex. While assembly activations were often correlated with movement or rest periods (*Figure 9—figure supplement 1*), and 60% of assemblies in both the peri-infarct and distal regions were significantly correlated with animal movement at all time points (*Figure 9—figure supplement 1C and E*, respectively), stroke did not change the average correlation between assembly activation and animal movement in either the peri-infarct or distal region (*Figure 9—figure supplement 1B and D*, respectively).

## Discussion

Here, we used longitudinal two photon calcium imaging of awake, head-fixed mice in a mobile homecage to examine how focal photothrombotic stroke to the forelimb sensorimotor cortex alters the activity and connectivity of neurons adjacent and distal to the infarct. Consistent with previous studies using intrinsic optical signal imaging, mesoscale imaging of regional calcium responses

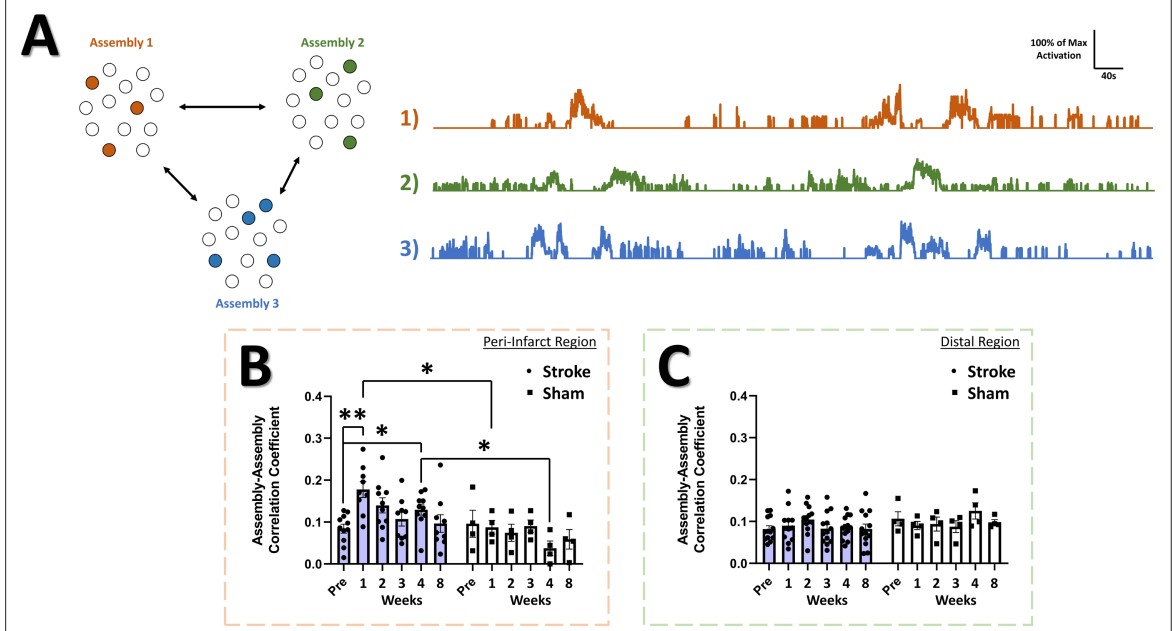

**Figure 9.** Correlation among neural assemblies is increased in the peri-infarct imaging region after stroke. (**A**) Illustration of three example color coded neural assemblies from a neural population and their corresponding time series activations that were correlated. In the peri-infarct imaging region, a significant main effect of group was found in the correlation coefficient for significantly correlated assembly-assembly pairs (**B**), with *post-hoc* tests indicating a significantly higher correlation at 2 week (p=0.029) and 4 weeks (p=0.0253) in the stroke group relative to sham, and at 1 week (p=0.0084) and 4 week (p=0.0261) in the stroke group compared to pre-stroke. Mixed Effects Model, Time $F_{(3.718, 42.39)}$=2.014, p=0.1141; Stroke Group $F_{(1, 13)}$=9.155, p=0.0097; Interaction $F_{(5, 57)}$=2.348, p=0.0524. In the distal imaging region, no significant main effects or interaction was found for the correlation coefficient for significantly correlated assembly-assembly pairs (**C**). Peri-infarct region Stroke N=11, Sham N=4. Distal region Stroke N=13, Sham N=4. *p<0.05; **p<0.01; ***p<0.001.

The online version of this article includes the following figure supplement(s) for figure 9:

**Figure supplement 1.** Correlation between assembly activations and movement is not detectably affected by stroke.

(reflecting bulk neuronal spiking in that region) showed that targeted stroke to the cFL somatosensory area disrupts the sensory-evoked forelimb representation in the infarcted region. Consistent with previous studies, this functional representation exhibited a posterior shift 8 weeks after injury, with activation in a region lateral to the cHL representation. Notably, sensory-evoked cFL representations exhibited reduced amplitudes of activity relative to pre-stroke activation measured in the cFL representation and in the region lateral the cHL representation. Longitudinal two-photon calcium imaging in awake animals was used to probe single neuron and local network changes adjacent the infarct and in a distal region that corresponded to the shifted region of cFL activation. This imaging revealed a decrease in firing rate at 1 week post-stroke in the peri-infarct region that was significantly negatively correlated with the number of errors made with the stroke-affected limbs on the tapered beam task. Peri-infarct cortical networks also exhibited a reduction in the number of functional connections per neuron and a sustained disruption in neural assembly structure, including a reduction in the number of assemblies and an increased recruitment of neurons into functional assemblies. Elevated correlation between assemblies within the peri-infarct region peaked 1 week after stroke and was sustained throughout recovery. Surprisingly, distal networks, even in the region associated with the shifted cFL functional map in anaesthetized preparations, were largely undisturbed.

## Cortical plasticity after stroke

Plasticity within and between cortical regions contributes to partial recovery of function and is proportional to both the extent of damage, as well as the form and quantity of rehabilitative therapy post-stroke (*Murphy and Corbett, 2009*; *Xu et al., 2009*). A critical period of highest plasticity begins shortly after the onset of stroke, is greatest during the first few weeks, and progressively diminishes over the weeks to months after stroke (*Brown et al., 2009*; *Biernaskie et al., 2004*; *Brown*

*et al., 2007*; *Carmichael et al., 2005*; *Cheatwood et al., 2008*; *Ploughman et al., 2009*). Functional recovery after stroke is thought to depend largely on the adaptive plasticity of surviving neurons that reinforce existing connections and/or replace the function of lost networks (*Winship and Murphy, 2008*; *Winship and Murphy, 2009*; *Carmichael, 2003b*; *Carmichael, 2006*; *Dancause et al., 2005*). This neuronal plasticity is believed to lead to topographical shifts in somatosensory functional maps to adjacent areas of the cortex. The driver for this process has largely been ascribed to a complex cascade of intra- and extra-cellular signaling that ultimately leads to plastic re-organization of the microarchitecture and function of surviving peri-infarct tissue (*Winship and Murphy, 2009*; *Murphy and Corbett, 2009*; *Carmichael et al., 2005*; *Carmichael, 2006*; *Carmichael, 2003a*; *Li and Carmichael, 2006*; *Li et al., 2010*). Likewise, structural and functional remodeling has previously been found to be dependent on the distance from the stroke core, with closer tissue undergoing greater re-organization than more distant tissue (for review, see *Winship and Murphy, 2009*).

Previous research examining the region at the border between the cFL and cHL somatosensory maps has shown this region to be a primary site for functional remapping after cFL directed photothrombotic stroke, resulting in a region of cFL and cHL map functional overlap (*Winship and Murphy, 2008*). Within this overlapping area, neurons have been shown to lose limb selectivity 1 month post-stroke (*Winship and Murphy, 2008*). This is followed by the acquisition of more selective responses 2 months post-stroke and is associated with reduced regional overlap between cFL and cHL functional maps (*Winship and Murphy, 2008*). Notably, this functional plasticity at the cellular level was assessed using strong vibrotactile stimulation of the limbs in anaesthetized animals. Our findings using longitudinal imaging in awake animals show an initial reduction in firing rate at 1 week post-stroke within the peri-infarct region that was predictive of functional impairment in the tapered beam task. This transient reduction may be associated with reduced or dysfunctional thalamic connectivity (*Paz et al., 2010*; *Staines et al., 2002a*; *Thomas Carmichael et al., 2001*) and reduced transmission of signals from hypo-excitable thalamo-cortical projections (*Tennant et al., 2017*). Importantly, the strong negative correlation we observed between firing rate of the neural population within the peri-infarct cortex and the number of errors on the affected side, as well as the rapid recovery of firing rate and tapered beam performance, suggests that neuronal activity within the peri-infarct region contributes to the impairment and recovery. The common timescale of neuronal and functional recovery also coincides with angiogenesis and re-establishment of vascular support for peri-infarct tissue (*Brown et al., 2007*; *Brown and Murphy, 2008*; *Ergul et al., 2012*; *Greenberg, 2014*; *Hatakeyama et al., 2020*).

Consistent with previous research using mechanical limb stimulation under anaesthesia (*Winship and Murphy, 2008*), we show that at the 8-week timepoint after cFL photothrombotic stroke the cFL representation is shifted posterior from its pre-stroke location into the area lateral to the cHL map. Notably, our distal region for awake imaging was directly within this 8-week post-stroke cFL representation. Despite our prediction that this distal area would be a hotspot for plastic changes, there was no detectable alteration to the level of population correlation, functional connectivity, assembly architecture or assembly activations after stroke. Moreover, we found little change in the firing rate in either moving or resting states in this region. Contrary to our results, somatosensory-evoked activity assessed by two photon calcium imaging in anesthetized animals has demonstrated an increase in cFL responsive neurons within a region lateral to the cHL representation 1–2 months after focal cFL stroke (*Winship and Murphy, 2008*). Notably, this previous study measured sensory-evoked single cell activity using strong vibrotactile (1 s 100 Hz) limb stimulation under aneasthesia (*Winship and Murphy, 2008*). This frequency of limb stimulation has been shown to elicit near maximal neuronal responses within the limb-associated somatosensory cortex under anesthesia (*Bandet et al., 2021*). Thus, strong stimulation and anaesthesia may have unmasked non-physiological activity in neurons in the distal region that is not apparent during more naturalistic activation during awake locomotion or rest. Regional mapping defined using strong stimulation in anesthetized animals may therefore overestimate plasticity at the cellular level.

Our results suggest a limited spatial distance over which the peri-infarct somatosensory cortex displays significant network functional deficits during movement and rest. Our results are consistent with a spatial gradient of plasticity mediating factors that are generally enhanced with closer proximity to the infarct core (*Carmichael et al., 2005*; *Carmichael, 2006*; *Carmichael, 2003a*; *Li and Carmichael, 2006*). However, our analysis outside peri-infarct cortex is limited to a single distal area caudal to the pre-stroke cFL representation. Although somatosensory maps in the present study were

defined by a statistical criterion for delineating highly responsive cortical regions from those with weak responses, the distal area in this study may have been a site of activity that did not meet the statistical criterion for inclusion in the baseline map. The lack of detectable changes in population correlations, functional connectivity, assembly architecture and assembly activations in the distal region may reflect minimal pressure for plastic change as networks in regions below the threshold for regional map inclusion prior to stroke may still be functional in the distal cortex. Thus, threshold-based assessment of remapping may further overestimate the neuroplasticity underlying functional reorganization suggested by anaesthetized preparations with strong stimulation. Future studies could examine distal areas medial and anterior to the cFL somatosensory area, such as the motor and pre-motor cortex, to further define the effect of FL targeted stroke on neuroplasticity within other functionally relevant regions. Moreover, the restriction of these network changes to peri-infarct cortex could also reflect the small penumbra associated with photothrombotic stroke, and future studies could make use of stroke models with larger penumbral regions, such as the middle cerebral artery occlusion model. Larger injuries induce more sustained sensorimotor impairment, and the relationship between neuronal firing, connectivity, and neuronal assemblies could be further probed relative to recovery or sustained impairment in these models. Recent research also indicates that stroke causes distinct patterns of disruption to the network topology of excitatory and inhibitory cells (*Latifi et al., 2020*), and that stroke can disproportionately disrupt the function of high activity compared to low activity neurons in specific neuron sub-types (*Motaharinia et al., 2021*). Mouse models with genetically labeled neuronal sub-types (including different classes of inhibitory interneurons) could be used to track the function of those populations over time in awake animals. A potential limitation of our data is the undefined effect of age and sex on cortical dynamics in this cohort of mice (with ages ranging from 3 to 9 months) after stroke. Aging can impair neurovascular coupling (*Zaletel et al., 2005*; *Akif Topcuoglu et al., 2009*; *Stefanova et al., 2013*; *Fabiani et al., 2014*; *Sorond et al., 2013*; *Park et al., 2007*) and reduce ischemic tolerance (*Menon et al., 2013*; *Faber et al., 2011*; *Strandgaard, 1991*; *Ay et al., 2005*), and greater investigation of cortical activity changes after stroke in aged animals would more effectively model stroke in humans. Future research could replicate this study with mice in middle-age and aged mice (e.g. 9 months and 18+months of age), and with sufficient quantities of both sexes, to better examine age and sex effects on measures of cortical function.

## Altered network connectivity after stroke

A highly distributed network of neural circuits forms the basis of information flow in the mammalian brain (*Bullmore and Sporns, 2009*; *Sporns, 2011*). Many disease states are thought to result in disturbances in neural dynamics and connectivity of these networks due to a process of randomization that affects the network nodes and connections, leading to degraded functional performance of the networks (*Sporns, 2011*; *Stam, 2014*). Most studies looking at the functional connectivity of cortical networks after stroke have focused on connectivity between cortical regions and few have looked at functional connectivity within neural populations at the single neuron level within a region of cortex. Our finding of decreased number of functional connections per neuron in the peri-infarct somatosensory cortex are consistent with recent research that demonstrated a decrease in the total number of functional connections for both inhibitory cells (*Latifi et al., 2020*) and excitatory cells (*Latifi et al., 2020*; *Bechay et al., 2022*) in the motor and premotor cortex approximately 1 week after focal sensorimotor stroke. Reductions in strong functional connections between neurons in our experiments peaked 1 week after stroke, with only partial recovery over the next 7 weeks. These changes were restricted to peri-infarct cortex and were not observed in distal regions. As the cumulative total number of functional connections within a neural network is dependent on the number of cells measured from the population, it is possible that a decrease in the number of connections at early timepoints in both studies may either reflect a loss of neurons and/or the functional silencing of neurons within the imaged areas. Contrary to the findings from *Bechay et al., 2022*, connection density (as % of max possible connections) did not decrease in either the peri-infarct somatosensory cortex or distal somatosensory cortex after stroke in our study. Thus, we observed a significant reduction in the number of strong functional connections between neurons, but not a reduction in the density of connections. It is notable that the peak deficit in peri-infarct functional connectivity in our study occurs simultaneously with reduced neuronal firing and sensorimotor impairment. Early disruption has been reported in anaesthetized animals, and our data in awake animals demonstrates

that early network changes may be a key contributor to dysfunction, and a restoration of spiking in peri-infarct regions is related to recovery (*Brown et al., 2009*; *Chen et al., 2012*; *Lim et al., 2014*; *Sigler et al., 2009*; *Sweetnam and Brown, 2013*; *Sweetnam et al., 2012*; *Winship and Murphy, 2008*).

## Altered network dynamics after stroke

According to the cell assembly hypothesis, transient synchronous activation of distributed groups of neurons organize into "neural assemblies" that underly the representation of both external and internal stimuli in the brain (*Buzsáki, 2010*). These assemblies have been demonstrated in the mammalian (*Berkes et al., 2011*; *Cossart et al., 2003*; *Harris et al., 2003*; *Hyman et al., 2013*; *Malvache et al., 2016*; *Miller et al., 2014*; *Sakata and Harris, 2009*; *See et al., 2018*; *Truccolo et al., 2010*; *Villette et al., 2015*) and zebrafish (*Romano et al., 2015*; *Avitan et al., 2017*; *Pietri et al., 2017*; *Thompson and Scott, 2016*) brain, and are often similar in presentation between spontaneous and sensory evoked activations (*Romano et al., 2015*; *Berkes et al., 2011*; *Miller et al., 2014*; *Luczak et al., 2009*; *MacLean et al., 2005*). Indeed, the similarity between spontaneous assemblies and sensory evoked assemblies tends to increase during development (*Berkes et al., 2011*), potentially illustrating the 'sculpting' of neural circuits towards commonly encountered sensory stimuli. To date, however, no study has examined changes to the architecture of neural assemblies on the level of single cells within neural networks of the somatosensory cortex in the post-stroke brain. Our data shows a transient decrease in the number of assemblies and an increase in the number of neurons per assembly for the peri-infarct cortex after stroke. If assemblies within peri-infarct somatosensory cortex contribute to differential processing of distinct elements of sensory experience, a loss in the number of assemblies may contribute to a loss in sensory range, as is observed in human stroke patients (*Carey and Matyas, 2011*; *Connell et al., 2008*; *Kwakkel et al., 2004*; *Patel et al., 2000*; *Tyson et al., 2008*). Although there have been calculations on the typical size of cell assemblies in several regions of the brain as a percentage of the network population (for review, see *Buzsáki, 2010*), whether there is an optimal number of assemblies or optimal membership size of a neural assembly for a particular neural population is currently unknown. It is possible that a decrease in the number of assemblies and an increase in the membership of neurons in remaining assemblies may signify inefficient or ineffective forms of sensory processing. Likewise, it has been shown that electrical stimulation of single somatosensory neurons is sufficient to bias animal behavior towards a desired behavioral response (*Brecht et al., 2004*; *Houweling and Brecht, 2008*), thereby suggesting that a sparse neural code may be sufficient for sensation. It has also been argued that minimizing correlation within the neural population serves to reduce representational redundancy and improves representational efficiency (*Simoncelli and Olshausen, 2001*; *Barlow, 2001*), thereby suggesting that fewer assemblies, with each containing more members, may be an inefficient form of representational coding for sensory information. Furthermore, we observed a persistent increase in the correlation between the activations of assemblies in the peri-infarct cortex that peaks 1 week post-stroke. Co-activation of multiple assemblies simultaneously may decrease the signal-to-noise ratio of the representational information that each specific assembly holds. It may further reduce efficient information transfer within and between cortical networks, and may relate to prolonged modes of somatosensory activation (*Brown et al., 2009*) and reduced fidelity in response to sensory stimuli observed in mouse models of focal FL stroke (*Sweetnam and Brown, 2013*).

In summary, this work demonstrates that the peri-infarct region near to the stroke core displays decreased neuronal activity 1 week after stroke that is predictive of poor performance on the tapered beam task. Moreover, local neuronal networks immediately adjacent the stroke exhibited reductions in functional connectivity and sustained alterations in the structure of neuronal assemblies, suggesting that focal injury leads to long lasting alterations in functional properties in local neuronal networks. Interestingly, altered connectivity and neuronal assembly structure was not detected in more distal cortical regions despite these regions being located within regions of cortex that appear to undergo functional remapping measured by cortical activation in response to vibrotactile limb stimulation in anaesthetized animals. The persistence of these distal networks during recovery highlights the importance of awake imaging and naturalistic stimuli to fully define functional network changes during recovery from nervous system injury.

## Materials and methods

### Animals

Three to nine month old Thy1-GCaMP6S mice (Jackson Labs GP4.3, Strain #024275), N=16 stroke (average age: 5.4 months; 13 male, 3 female), and 5 sham (average age: 6 months; 3 male, 2 female), were used in this study. Mice were group housed in standard laboratory cages in a temperature-controlled room (23 °C), maintained on a 12 hr light/dark cycle, and given standard laboratory diet and water ad libitum. Cages were randomly assigned to stroke or sham groups such that stroke and sham mice did not co-habitate the same cages. Animal weight was monitored daily during the entirety of the experiment. All experiments were approved by the University of Alberta's Health Sciences Animal Care and Use Committee (Protocol AUP361) and adhered to the guidelines set by the Canadian Council for Animal Care. Animals were euthanized through decapitation under deep urethane anesthesia following the end of the final imaging session.

### Chronic cranial window implantation

Mice were implanted with a chronic cranial window (*Holtmaat et al., 2009*) 4 weeks prior to the first imaging time point (*Figure 1B*). A surgical plane of anesthesia was achieved with 1.5% isoflourane. Body temperature was measured using a rectal probe and maintained at 37 ± 0.5°C. Mice were administered 0.15 mL of saline subcutaneously to maintain hydration levels. Dexamethasone (2 μg/g) was given subcutaneously to prevent cortical swelling and hemorrhaging during the craniotomy procedure. The skull was exposed by midline scalp incision and the skin retracted. The skull was gently scraped with a scalpel to remove the periosteum. Grooves were scraped into the skull surface to improve adhesion of dental cement to the skull. A 4x4 mm region of the skull overlying the right hemisphere somatosensory region was thinned to 25–50% of original thickness using a high-speed dental drill (~1–5 mm lateral,+2 to –2 mm posterior to bregma). A dental drill was used to progressively thin the overlying skull until the bone could be removed with forceps, leaving the dura intact. The exposed cortical surface was bathed in sterile saline solution. A 5 mm diameter coverslip was held in place over the craniotomy and its edges attached to the skull using cyanoacrylate glue. A metal headplate was positioned and secured to the skull using dental cement. Animals were injected subcutaneously with buprenorphine (1.0 mg/kg), removed from the isofluorane, and allowed to recover in a temperature-controlled recovery cage. Mice were returned to their home cage once recovered and monitored daily for weight and post-surgical signs for the duration of the chronic experiment. Mice were allowed to recover for a period of 2 weeks after window implantation prior to beginning behavior and task habituation (*Figure 1C*). Animals were excluded at the 2 week post-implantation timepoint if their cranial window became cloudy or non-imageable.

### Tapered beam task habituation, testing and analysis

Methods for tapered beam habituation and automated recording and analysis have been described previously (*Ardesch et al., 2017*), with minor modification as follows. Nesting material from the homecage was placed inside of a dark box at the narrow end of the beam to motivate animals to cross the beam. For the first three sessions, mice were placed at the wide end of the beam and allowed to freely explore the beam for a period of 2 min, after which the experimenter placed a sheet of paper behind the animal as it crossed the beam to block its return path to the wide end and motivate it to cross to the narrow end with dark box. Once the mouse had reached the dark box with bedding material at the narrow end, they were given 60 s within the dark box to associate crossing the beam with reaching the safety of the dark box. The dark box was transported to their respective cage for the animal to further associate reaching the dark box with a safe transition to their cage. On subsequent days, mice were continuously run through the tapered beam for a period of 2 min each, with a return to their cage after each crossing. On each of the 3 days prior to the first Ca²⁺ imaging session, mice were tested with three crossings of the beam per day to determine baseline crossing performance. Left and right-side slips were captured automatically by Raspberry Pi computer attached to touch sensors on the tapered beam, and Python scripts run to determine number of slips for each side and distance to first slip. Performance on the 9 baseline trials were average to determine a singular average baseline performance level. Mice were tested with three trials on each day prior to each weekly post- imaging timepoint to measure changes in performance.

## String pull task habituation, testing and analysis

Methods for string pull habituation and testing have previously been described (*Blackwell et al., 2018a*; *Inayat et al., 2020*), with modification as follows. For habituation, mice were individually placed in a transparent rectangular cage without bedding and allowed to freely explore for a period of 5 min. Twenty strings with variable length were hung over the edge of the cage. Half of the strings were baited with chocolate flavored sucrose pellets. Mice were removed from the apparatus once all of the strings had been pulled into the cage, or once 20 min had elapsed. On subsequent days 2 and 3, mice were again placed in the cage with 20 strings to pull. Following this 3 day habituation, from days 4 to 14 mice were habituated in an alternate transparent string pull box with high sides and transparent front face for video recording. Within this second habituation box, mice received three trials with baited strings hung facing the recording camera. The session was terminated when the animal had pulled all three strings or once 20 min had elapsed, and the apparatus prepared for the next mouse. On the last day prior to the first $Ca^{2+}$ imaging session, three trials of string pull from each mouse were recorded with the use of a GoPro Hero 7 Black (60 fps, 1920×1080 pixels). Mice were re-tested with this same three-string protocol the day prior to each of their weekly imaging time-points. Video recordings were analyzed using a semi-automated Matlab string pull package (*Inayat et al., 2020*). Reach and withdraw movement scaling for the left and right paw was calculated by running a Pearson's correlation between the series of Euclidian distances that the paw travelled during each of the reach or withdraw movements during the string pull and correlating with the peak speed of the paw during each of the respective reaches or withdraws (*Blackwell et al., 2018b*). A value of 1 indicates a strong linear relationship between longer reach/withdraw paw movement distance and greater peak speed that the paw obtained. Paw reach distance was calculated as the average distance that the paw travelled during each reach movement. Path circuity is used as a measure of how direct of a path a reach or withdraw movement takes between its starting and end points and is calculated as the total distance travelled divided by the Euclidian distance between the start and end points (*Blackwell et al., 2018b*). The greater the path circuity value is above 1, the more the path of the paw deviated from the ideal direct path between the start and end points of the reach/withdraw. A bimanual correlation coefficient was calculated by first detrending the timeseries of Y-axis paw position to remove general changes in the posture of the animal over the course of the string pull attempt, then running a Pearson's correlation between the Y-axis paw position measurements for the right and left paw.

## Mobile floating homecage habituation and measurement of movement parameters

Methods for floating homecage (Neurotar, Finland) habituation have previously been described (*Kislin et al., 2014*), with modification as follows. Two weeks after cranial window implantation, each mouse was handled for 5 min three times per day for the first 2 days to habituate them to handling. For days 3–4, each mouse was handled and repeatedly wrapped and released with a soft cloth for a period of 5 min three times per day in order to habituate to the wrapping procedure. Mice were also given 1 period of 5 min each per day to freely explore the floating homecage without being head-restrained. For days 5–8, twice per day the animals were head-restrained in the floating homecage and allowed to move around the floating homecage for a period of 15 min with the room lights off before being returned to their cage. For days 8–14, twice per day the mice were head-restrained in the floating homecage for 25 min with the floating homecage attached to the microscope in the same conditions in which the animals would be imaged on day 15 onwards. On day 14 (one day prior to their first imaging session) baseline performance on the string pull and tapered beam tasks was measured prior to their last habituation in the floating homecage. Animal movement within the homecage during each $Ca^{2+}$ imaging sessions was tracked to determine animal speed and position. Movement periods were manually annotated on a subset of timeseries by co-recording animal movement using both the Mobile Homecage tracker, as well as a webcam (Logitech C270) with infrared filter removed. Movement tracking data was low pass filtered to remove spurious movement artifacts lasting below 6 recording frames (240ms). Based on annotated times of animal movement from the webcam recordings and Homecage tracking, a threshold of 30 mm/s from the tracking data was determined as frames of animal movement, whereas speeds below 30 mm/s were taken as periods of rest.

## Identification of forelimb and hindlimb somatosensory cortex representations using widefield Ca²⁺ imaging of sensory-evoked responses

On day 10 of habituation, mice were anesthetized with 1.25% isofluorane after they had completed all daily task training and floating homecage habituation. Mice were head-fixed on a stereotaxic frame with continuous isofluorane anesthesia. Body temperature was measured using a rectal probe and maintained at 37 ± 0.5°C. The cortical surface was illuminated with blue light from a xenon lamp (excitation band-pass filtered at 450–490 nm). Fluorescent emissions were long-pass filtered at 515 nm and captured in 12-bit format by a Dalsa Pantera 1M60 camera mounted on a Leica SP5 confocal microscope using a 2.5 X objective. The depth of focus was set at 200 µm below the cortical surface. Custom-made piezoceramic mechanical bending actuators were used to elicit oscillatory limb stimulation during widefield Ca²⁺ imaging. The piezo bending actuators comprised a piezo element (Piezo Systems # Q220-A4-203YB) attached to an electrically insulated metal shaft holding a metal U-bend at its end. The palm of the mouse paw was placed within the U-bend, and the U-bend bent to shape to lightly secure the palm. The metal U-bend made contact across a vertical rectangular area of approximately 3x1 mm on the palmar and dorsal surface of the hand. Stimulators were driven with square-wave signals from an A-M Systems Model 2100 Isolated Pulse Stimulator. Stimulation alternated between contralateral forelimb (cFL) and contralateral hindlimb (cHL) for up to 40 trials of stimulation of each limb. Placement of actuators was on the glabrous skin of the forepaw or hindpaw, with consistent alignment relative to the flexion of wrist and ankle. Images were captured for 5.0 s at 10 Hz (1 s before and 4 s after stimulus onset; interstimulus interval = 20 s). The trials for each limb were averaged in ImageJ software (NIH). 10 imaging frames (1 s) after stimulus onset were averaged and divided by the 10 baseline frames 1 s before stimulus onset to generate a response map for each limb. Response maps were thresholded at 5 times the standard deviation of the baseline period deltaFoF to determine limb associated response maps. These were merged and overlaid on an image of surface vasculature to delineate the cFL and cHL somatosensory representations and were also used to determine peak Ca²⁺ response amplitude from the timeseries recordings. For cFL stimulation trials, an additional ROI was placed over the region lateral to the cHL representation (denoted as 'distal region' in *Figure 2E*) to measure the distal region cFL evoked Ca²⁺ response amplitude pre- and post-stroke. The dimensions and position of the distal ROI was held consistent relative to surface vasculature for each animal from pre- to post-stroke.

## Calcium imaging

On the day following testing for behavioral performance on the string pull and tapered beam tasks, animals were head-fixed within the mobile homecage and Ca²⁺ imaging was performed. Awake in-vivo two-photon imaging was performed using a Leica SP5 MP confocal microscope equipped with a Ti:Sapphire Coherent Chameleon Vision II laser tuned to 940 nm for GCaMP6S excitation. A Leica HCX PL APO L 20x1.0 NA water immersion objective was used. Images were acquired using Leica LAS AF using no averaging, a zoom of 1.7 x and a frame-rate of 25 Hz. Images were acquired at 512x512 pixels over an area of 434x434 µm, yielding a resolution of 0.848 µm per pixel. Two imaging regions were chosen for each animal, with one corresponding to the half-way point between the pre-stroke cFL and cHL somatosensory maps previously determined with widefield Ca²⁺ imaging (the 'peri-infarct' imaging region), and the other an imaging region located just lateral to the cHL sensory map (the 'distal' imaging region). These regions can be seen in *Figure 1D* and *Figure 2A*. Each imaging region was recorded for 15 minutes while simultaneously tracking the animal's movement and position within the mobile homecage. Imaging depth was set between 100–180 µm below the cortical surface. There were no differences between the stroke group and sham group for movement within the mobile homecage, though both groups displayed an equivalent decrease in the amount of time they spent moving within the 15-min recording sessions over the course of the 8-week experimental timeline, potentially reflecting continuing habituation over this period.

## Photothrombotic stroke and sham procedure

After baseline cellular Ca²⁺ imaging, mice were anesthetized using 1.25% isofluorane and head-fixed on a stereotaxic frame with continuous isofluorane anesthesia. Body temperature was measured using a rectal probe and maintained at 37 ± 0.5°C. The mapped cFL area was used as a guide for a targeted

photothrombosis procedure. Briefly, a 2 mm diameter hole was punched in black electrical tape and attached onto the glass window to block stray light from illuminating cortical areas outside of the desired region. Rose Bengal (a photosensitive dye) was dissolved in 0.01 M sterile phosphate buffered saline (Sigma) and injected intraperitoneal (30 mg/kg). The cFL cortical area visible through the punched electrical tape was illuminated using a collimated beam of green laser light (532 nm, 17 mW) for 20 min to photoactivate the Rose Bengal and cause a focal ischemic lesion. Sham controls were treated in the same manner as stroke, however illumination of the laser was omitted.

## Two-photon image processing and analysis of neuronal activity

Timeseries images (15 min per imaging region, 22,500 frames per recording, 25fps) were group averaged in groups of 2, then motion corrected using the TurboReg plugin in FiJi with translation only registration (*Schindelin et al., 2012*). Z-projections within FiJi of the average and standard deviation were used to define neuronal ROI through manual tracing. Using custom written scripts in Matlab 2020b, ROIs were imported and converted into a format suitable for the $Ca^{2+}$ imaging toolbox used for subsequent steps (*Romano et al., 2017*). Neuropil for each ROI was determined by expanding an annular donut around each ROI to calculate the neuropil deltaF/Fo surrounding each ROI. During computation of neuron ROI deltaF/Fo, fluorescence was corrected for neuropil contamination using the formula ($F_{corrected} = F_{raw} - \alpha * F_{neuropil}$) where $\alpha$ was set to 0.4 based on previous studies suggesting that $\alpha$ values between 0.3–0.5 were optimal (*Peron et al., 2015*), and with $F_{neuropil}$ determined by the peri-somatic donut neuropil fluorescence closely associated with each neuron. A smoothed estimate of the baseline fluorescence was calculated by taking a 30 s running average of the 8th percentile of the raw fluorescence, which was then subtracted from the raw fluorescence to remove baseline drift.

## $Ca^{2+}$ trace deconvolution to determine neuronal transients

To determine significant fluorescent $Ca^{2+}$ transients from the deltaF/Fo of each neuron, a dynamic threshold implementing a Bayesian odds ratio estimation framework for noise estimation was used to determine transients that met the condition of being greater than 98% of the confidence interval for the calculated fluorescent baseline noise, and were compatible with a tau of ~1.8 s for GCaMP6S (*Romano et al., 2015*; *Romano et al., 2017*). Determination of firing was performed on the neuronal deltaF/Fo traces, as implemented in *Romano et al., 2015*; *Romano et al., 2017*, in order to define the $Ca^{2+}$ transient rate and to generate raster plots. This information was used to determine neuronal activity 'firing rate' as shown in *Figure 4*.

## Neuron-neuron correlation analysis and functional connectivity

Using custom written scripts in Matlab, Pearson product-moment correlation coefficients were calculated between the z-scored $Ca^{2+}$ traces derived from each of the neuronal ROIs in a given optical section. Distance between neuron pairs was calculated from the Euclidean distance between the central points of the neuron ROIs of the image frame. Functional connectivity plots (*Figure 5*) were generated by plotting neuron ROI centroids as red dots, and lines between them with line weight and color determined by the strength of the correlation between them. Stationary variable block bootstrapping (5000 iterations) was performed as a statistical test for the significance of each pairwise correlation and only correlations that were greater than the 99th percentile of the bootstrap were deemed statistically significant and plotted. Average number of significant connections per neuron was calculated by dividing the number of functional connections that met the bootstrapping threshold by the total number of neurons in the population. Connection density was calculated as percent of max by taking the total number of significant functional connections and dividing by the total potential number of functional connections if every neuron within the population was functionally connected with all other neurons.

## Determination of neural assemblies and their activity patterns

To evaluate the co-activity patterns of neurons that form putative neural assemblies, a PCA-Promax procedure was applied as previously described (*Romano et al., 2015*; *Romano et al., 2017*), with a zMax threshold manually selected by the first clear minimum in the distribution of z-scored maximal ROI loadings. Notably, the PCA-Promax procedure relaxes the PCA orthogonality condition using a PROMAX oblique rotation of the PC axes (*Hendrickson and White, 1964*), such that assemblies can

contain neurons found within other assemblies. Due to this relaxation of the PCA orthogonality condition, assemblies with high overlap whose dot product exceeded 0.6 were merged (*Hendrickson and White, 1964*). All assemblies were compared to surrogate control datasets, and only those assemblies whose members were significantly correlated and synchronous were kept (p<0.05). To determine the co-activity of neurons within assemblies, a matching index was calculated (*Romano et al., 2015*; *Romano et al., 2017*; *Hilgetag et al., 2002*; *Sporns et al., 2007*). The matching index quantifies the proportion of neurons within the assembly that are co-activated simultaneously over the timeseries, with a maximal value of 1 indicating perfect overlap in assembly member activation. The significance of each assembly activation over the timeseries was determined by comparing the activation events to a probability distribution based on the size of the assembly, the size of the total population of N ROIs, and with a threshold p-value of <0.05 considered to be a statistically significant activation. For cross-correlation of assembly-assembly correlations (*Figure 9*) and assembly activations with speed (*Figure 9—figure supplement 1*), the matching index was first multiplied by the Boolean timeseries of individual assembly activation significance to generate matching index timeseries of significant assembly activations only (presented in *Figure 9* and *Figure 9—figure supplement 1* as percent of maximal assembly activation). ROIs of assembly members were color coded according to their assembly membership and overlaid on an averaged fluorescence image to generate assembly plots (*Figure 7*).

## Statistical analysis

Multivariate comparisons were made using a mixed-effects model for repeated measures based on a restricted maximum likelihood generalized linear mixed model as implemented in Graphpad Prism 9.0.0, with Bonferroni-Sidak corrections used for *post-hoc* comparisons. Bonferroni-Sidak post hoc testing was used to compare the means of the stroke vs. the sham group at each timepoint, as well as to identify within group differences by comparing different timepoints within each group. Grubb's test was used to remove data outliers with alpha set to 0.05. All zero time-lag cross-correlations were computed using Pearson's *r*. A stationary bootstrapping procedure was used as a test of the significance of the calculated Pearson's r of each pairwise timeseries comparison. Within the stationary bootstrapping procedure, 5000 iterations were run and an average block length of 23 frames (1.84 s) was used based on previous studies indicating ~1.8 s as the decay time for GCaMP6S $Ca^{2+}$ sensor within neurons (*Chen et al., 2013*; *Dana et al., 2019*). Pairwise r values greater than the 99h percentile of the stationary bootstrap were deemed 'significant'. Non-significant pairwise r values were excluded from analyses of average correlation values. For all statistical comparisons, a p value of$\leq$0.05 was considered statistically significant. A p value of$\leq$0.10 was considered a statistical trend. Data are expressed as the mean ± SEM.

## Additional information

### Funding

| Funder | Grant reference number | Author |
| --- | --- | --- |
| Natural Sciences and Engineering Research Council of Canada | RGPIN-2023-04969 | Ian Robert Winship |
| Canadian Institutes of Health Research | PS 166144 | Ian Robert Winship |

The funders had no role in study design, data collection and interpretation, or the decision to submit the work for publication.

### Author contributions

Mischa Vance Bandet, Conceptualization, Data curation, Software, Formal analysis, Validation, Investigation, Visualization, Methodology, Writing - original draft, Writing - review and editing; Ian Robert Winship, Conceptualization, Resources, Supervision, Funding acquisition, Validation, Investigation, Methodology, Project administration, Writing - review and editing

## Author ORCIDs
Mischa Vance Bandet http://orcid.org/0000-0002-3744-441X
Ian Robert Winship http://orcid.org/0000-0002-8574-4855

## Ethics
All experiments were approved by the University of Alberta's Health Sciences Animal Care and Use Committee (Protocol AUP361) and adhered to the guidelines set by the Canadian Council for Animal Care.

Reviewer #1 (Public Review): https://doi.org/10.7554/eLife.90080.3.sa1
Reviewer #2 (Public Review): https://doi.org/10.7554/eLife.90080.3.sa2
Author response https://doi.org/10.7554/eLife.90080.3.sa3

## Additional files

### Supplementary files
• MDAR checklist

### Data availability
All data generated or analyzed during this study has been made publicly available on OSF (https://doi.org/10.17605/OSF.IO/WAM9J).

The following dataset was generated:

| Author(s) | Year | Dataset title | Dataset URL | Database and Identifier |
|---|---|---|---|---|
| Bandet MV, Winship IR | 2024 | Aberrant cortical activity, functional connectivity, and neural assembly architecture after photothrombotic stroke in mice | https://doi.org/10.17605/OSF.IO/WAM9J | Open Science Framework, 10.17605/OSF.IO/WAM9J |

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
