## [Editor Report · eLife assessment]

This **important** study sheds light on several apparent discrepancies observed across animal studies examining neuroimaging biomarkers of functional recovery following focal ischemia. Using 2-photon imaging of calcium activity in awake mice, the authors show **compelling** evidence that deficits in neuronal activity and functional connectivity after photothrombosis occur within a very small distance from the infarct (<750 microns) whereas these measures were relatively unaltered more distally, even those typically implicated with functional remapping of the forelimb representation in anaesthetized animals. These findings reveal a complex spatiotemporal relationship between perilesional neuronal network function and behavioral recovery that is more nuanced than previously reported, and motivates the need for better criteria for what is considered remapping.

---

## [Referee Report · Reviewer #1 (Public Review)]

Summary:

This impressive study by Bandet and Winship uses 2-photon imaging in awake behaving mice to examine long-term changes in neural activity and functional connectivity after focal ischemic stroke. The authors discover that there are long-lasting perturbations in neural activity and functional connectivity, specifically within peri-infarct cortex but not more distant cortical regions. Overall I thought the study provided important new findings that were supported by compelling data.

Strengths:

This is a technically challenging study and the experiments appear to be well done. The manuscript was written in a concise manner, and the figures were clearly presented. The analytic tools were rigorous and appropriate, leading to novel insights regarding neural activity patterns during movement or rest. The discovery of long-lasting impairments in neural activity/functional connectivity is important (and often overlooked) given that future stroke studies need to recognize what problems exist in order to properly rectify them. The authors also question the spatial extent to which functional changes occur after stroke, at least at the single cell level. Overall, I think this was a well-executed study whose primary conclusions were justified by the data presented.

Weaknesses:

I found very little in the way of weaknesses. The authors addressed my comments about the methodology, statistical analysis, normalization of data and discussion points about cortical plasticity during stroke recovery.

---

## [Referee Report · Reviewer #2 (Public Review)]

This study investigates the excitability of neurons in the peri-infarct cortex during recovery from ischemic stroke. The excitability of neurons in the peri-infarct cortex during stroke recovery has produced contradictory findings: some studies suggest hyper-excitability to direct-brain stimulation, while others indicate diminished responsiveness to physical stimuli. However, most studies have used anesthetized animals, which can disrupt cortical activity and functional connectivity. The present study used two-photon Ca2+ imaging after focal photothrombotic stroke to examine neural activity patterns in awake mice. The authors found reduced neuronal spiking in the peri-infarct cortex that was strongly correlated with motor performance deficits. Additionally, the authors found disruptions in neural activation, functional connectivity, and assembly architecture in the immediate peri-infarct region but not in the distal cortex regions.

The findings of this study are very important as they show that there is no measurable change in terms of neuronal activation and reorganization in distal regions of remapped cortical response areas after stroke.

---

## [Author Response]

The following is the authors’ response to the original reviews.

**Reviewer #1 (Recommendations For The Authors):**
(1) Methods, please state the sex of the mice.

This has now been added to the methods section:

“Three to nine month old Thy1-GCaMP6S mice (Strain GP4.3, Jax Labs), N=16 stroke (average age: 5.4 months; 13 male, 3 female), and 5 sham (average age: 6 months; 3 male, 2 female), were used in this study.”

(2) The analysis in Fig 3B-D, 4B-C, and 6A, B highlights the loss of limb function, firing rate, or connections at 1 week but this phenomenon is clearly persisting longer in some datasets (Fig. 3 and 6). Was there not a statistical difference at weeks 2,3,4,8 relative to "Pre-stroke" or were comparisons only made to equivalent time points in the sham group? Personally, I think it is useful to compare to "pre-stroke" which should be more reflective of that sample of animals than comparing to a different set of animals in the Sham group. A 1 sample t-test could be used in Fig 4 and 6 normalized data.

On further analysis of our datasets, normalization throughout the manuscript was unnecessary for proper depiction of results, and all normalized datasets have been replaced with nonnormalized datasets. All within group statistics are now indicated within the manuscript.

(3) Fig 4A shows a very striking change in activity that doesn't seem to be borne out with group comparisons. Since many neurons are quiet or show very little activity, did the authors ever consider subgrouping their analysis based on cells that show high activity levels (top 20 or 30% of cells) vs those that are inactive most of the time? Recent research has shown that the effects of stroke can have a disproportionate impact on these highly active cells versus the minimally active ones.

A qualitative analysis supports a loss of cells with high activity at the 1-week post-stroke timepoint, and examination of average firing rates at 1-week shows reductions in the animals with the highest average rates. However, we have not tracked responses within individual neurons or quantitatively analyzed the data by subdividing cells into groups based on their prestroke activity levels. We have amended the discussion of the manuscript with the following to highlight the previous data as it relates to our study:

“Recent research also indicates that stroke causes distinct patterns of disruption to the network topology of excitatory and inhibitory cells [73], and that stroke can disproportionately disrupt the function of high activity compared to low activity neurons in specific neuron sub-types [61]. Mouse models with genetically labelled neuronal sub-types (including different classes of inhibitory interneurons) could be used to track the function of those populations over time in awake animals.”

(4) Fig 4 shows normalized firing rates when moving and at rest but it would be interesting to know what the true difference in activity was in these 2 states. My assumption is that stroke reduces movement therefore one normalizes the data. The authors could consider putting non-normalized data in a Supp figure, or at least provide a rationale for not showing this, such as stating that movement output was significantly suppressed, hence the need for normalization.

On further analysis of our datasets, normalization throughout the manuscript was unnecessary for proper depiction of results, and all normalized datasets have been replaced with nonnormalized datasets.

(5) One thought for the discussion. The fact that the authors did not find any changes in "distant" cortex may be specific to the region they chose to sample (caudal FL cortex). It is possible that examining different "distant" regions could yield a different outcome. For example, one could argue that there may have been no reason for this area to "change" since it was responsive to FL stimuli before stroke. Further, since it was posterior to the stroke, thalamocortical projects should have been minimally disturbed.

We would like to thank the reviewer for this comment. We have amended the discussion with the following:

“Our results suggest a limited spatial distance over which the peri-infarct somatosensory cortex displays significant network functional deficits during movement and rest. Our results are consistent with a spatial gradient of plasticity mediating factors that are generally enhanced with closer proximity to the infarct core [84,88,90,91]. However, our analysis outside peri-infarct cortex is limited to a single distal area caudal to the pre-stroke cFL representation. Although somatosensory maps in the present study were defined by a statistical criterion for delineating highly responsive cortical regions from those with weak responses, the distal area in this study may have been a site of activity that did not meet the statistical criterion for inclusion in the baseline map. The lack of detectable changes in population correlations, functional connectivity, assembly architecture and assembly activations in the distal region may reflect minimal pressure for plastic change as networks in regions below the threshold for regional map inclusion prior to stroke may still be functional in the distal cortex. Thus, threshold-based assessment of remapping may further overestimate the neuroplasticity underlying functional reorganization suggested by anaesthetized preparations with strong stimulation. Future studies could examine distal areas medial and anterior to the cFL somatosensory area, such as the motor and pre-motor cortex, to further define the effect of FL targeted stroke on neuroplasticity within other functionally relevant regions. Moreover, the restriction of these network changes to peri-infarct cortex could also reflect the small penumbra associated with photothrombotic stroke, and future studies could make use of stroke models with larger penumbral regions, such as the middle cerebral artery occlusion model. Larger injuries induce more sustained sensorimotor impairment, and the relationship between neuronal firing, connectivity, and neuronal assemblies could be further probed relative to recovery or sustained impairment in these models.”

Minor comments:Line 129, I don't necessarily think the infarct shows "hyper-fluorescence", it just absorbs less white light (or reflects more light) than blood-rich neighbouring regions.

Sentence in the manuscript has been changed to:

“Resulting infarcts lesioned this region, and borders could be defined by a region of decreased light absorption 1 week post-stroke (Fig 1D, Top).”

Line 130-132: the authors refer to Fig 1D to show cellular changes but these cannot be seen from the images presented. Perhaps a supplementary zoomed-in image would be helpful.

As changes to the morphology of neurons are not one of the primary objectives of this study, and sampled resolution was not sufficiently high to clearly delineate the processes of neurons necessary for morphological assessment, we have amended the text as follows:

“Within the peri-infarct imaging region, cellular dysmorphia and swelling was visually apparent in some cells during two photon imaging 1-week after stroke, but recovered over the 2 month poststroke imaging timeframe (data not shown). These gross morphological changes were not visually apparent in the more distal imaging region lateral to the cHL.”

Lines 541-543, was there a rationale for defining movement as >30mm/s? Based on a statistical estimate of noise?

Text has been altered as follows:

“Animal movement within the homecage during each Ca2+ imaging session was tracked to determine animal speed and position. Movement periods were manually annotated on a subset of timeseries by co-recording animal movement using both the Mobile Homecage tracker, as well as a webcam (Logitech C270) with infrared filter removed. Movement tracking data was low pass filtered to remove spurious movement artifacts lasting below 6 recording frames (240ms). Based on annotated times of animal movement from the webcam recordings and Homecage tracking, a threshold of 30mm/s from the tracking data was determined as frames of animal movement, whereas speeds below 30mm/s was taken as periods of rest.”

Lines 191-195: Note that although the finding of reduced neural activity is in disagreement with a multi-unit recording study, it is consistent with other very recent single-cell Ca++ imaging data after stroke (PMID: 34172735 , 34671051).

Text has been altered as follows:

“These results indicate decreased neuronal spiking 1-week after stroke in regions immediately adjacent to the infarct, but not in distal regions, that is strongly related to sensorimotor impairment. This finding runs contrary to a previous report of increased spontaneous multi-unit activity as early as 3-7 days after focal photothrombotic stroke in the peri-infarct cortex [1], but is in agreement with recent single-cell calcium imaging data demonstrating reduced sensoryevoked activity in neurons within the peri-infarct cortex after stroke [60,61].”

Fig 7. I don't understand what the color code represents. Are these neurons belonging to the same assembly (or membership?).

That is correct, neurons with identical color code belong to the same assembly. The legend of Fig 7 has been modified as follows to make this more explicit:

“Fig 7. Color coded neural assembly plots depict altered neural assembly architecture after stroke in the peri-infarct region. (A) Representative cellular Ca2+ fluorescence images with neural assemblies color coded and overlaid for each timepoint. Neurons belonging to the same assembly have been pseudocolored with identical color. A loss in the number of neural assemblies after stroke in the peri-infarct region is visually apparent, along with a concurrent increase in the number of neurons for each remaining assembly. (B) Representative sham animal displays no visible change in the number of assemblies or number of neurons per assembly.”

**Reviewer #2 (Recommendations For The Authors):**
Materials and methodsIdentification of forelimb and hindlimb somatosensory cortex representations [...] Cortical response areas are calculated using a threshold of 95% peak activity within the trial. The threshold is presumably used to discriminate between the sensory-evoked response and collateral activation / less "relevant" response (noise). Since the peak intensity is lower after stroke, the "response" area is larger - lower main signal results in less noise exclusion. Predictably, areas that show a higher response before stroke than after are excluded from the response area before stroke and included after. While it is expected that the remapped areas will exhibit a lower response than the original and considering the absence of neuronal activity, assembly architecture, or functional connectivity in the "remapped" regions, a minimal criterion for remapping should be to exhibit higher activation than before stroke. Please use a different criterion to map the cortical response area after stroke.

We would like to thank the reviewer for this comment. We agree with the reviewer’s assessment of 95% of peak as an arbitrary criterion of mapped areas. To exclude noise from the analysis of mapped regions, a new statistical criterion of 5X the standard deviation of the baseline period was used to determine the threshold to use to define each response map. These maps were used to determine the peak intensity of the forelimb response. We also measured a separate ROI specifically overlapping the distal region, lateral to the hindlimb map, to determine specific changes to widefield Ca2+ responses within this distal region. We have amended the text as follows and have altered Figure 2 with new data generated from our new criterion for cortical mapping.

“The trials for each limb were averaged in ImageJ software (NIH). 10 imaging frames (1s) after stimulus onset were averaged and divided by the 10 baseline frames 1s before stimulus onset to generate a response map for each limb. Response maps were thresholded at 5 times the standard deviation of the baseline period deltaFoF to determine limb associated response maps. These were merged and overlaid on an image of surface vasculature to delineate the cFL and cHL somatosensory representations and were also used to determine peak Ca2+ response amplitude from the timeseries recordings. For cFL stimulation trials, an additional ROI was placed over the region lateral to the cHL representation (denoted as “distal region” in Fig 2E) to measure the distal region cFL evoked Ca2+ response amplitude pre- and post-stroke. The dimensions and position of the distal ROI was held consistent relative to surface vasculature for each animal from pre- to post-stroke.”

AnimalsMice used have an age that goes from 3 to 9 months. This is a big difference given that literature on healthy aging reports changes in neurovascular coupling starting from 8-9 months old mice. Consider adding age as a covariate in the analysis.

We do not have sufficient numbers of animals within this study to examine the effect of age on the results observed herein. We have amended the discussion with the following to address this point:

“A potential limitation of our data is the undefined effect of age and sex on cortical dynamics in this cohort of mice (with ages ranging from 3-9 months) after stroke. Aging can impair neurovascular coupling [102–107] and reduce ischemic tolerance [108–111], and greater investigation of cortical activity changes after stroke in aged animals would more effectively model stroke in humans. Future research could replicate this study with mice in middle-age and aged mice (e.g. 9 months and 18+ months of age), and with sufficient quantities of both sexes, to better examine age and sex effects on measures of cortical function.”

StatisticsPlease describe the "normalization" that was applied to the firing rate. Since a mixedeffects model was used, why wasn't baseline simply added as a covariate? With this type of data, normalization is useful for visualization purposes.

On further analysis of our datasets, normalization throughout the manuscript was unnecessary for the visualization of results, and all normalized datasets have been replaced with nonnormalized datasets. All within group comparisons are now indicated throughout the manuscript and in the figures.

IntroductionLine 93 awake, freely behaving but head-fixed. That's not freely. Should just say behaving.

Sentence has been edited as follows:

“We used awake, behaving but head-fixed mice in a mobile homecage to longitudinally measure cortical activity, then used computational methods to assess functional connectivity and neural assembly architecture at baseline and each week for 2 months following stroke.”

110 - 112 The last part of this sentence is unjustified because these areas have been incorrectly identified as locations of representational remapping.

We agree with the reviewer and have amended the manuscript as follows after re-analyzing the dataset on widefield Ca2+ imaging of sensory-evoked responses:“Surprisingly, we also show that significant alterations in neuronal activity (firing rate), functional connectivity, and neural assembly architecture are absent within more distal regions of cortex as little as 750 µm from the stroke border, even in areas identified by regional functional imaging (under anaesthesia) as ‘remapped’ locations of sensory-evoked FL activity 8-weeks post-stroke.”

Results149-152 There is no observed increase in the evoked response area. There is an observed change in the criteria for what is considered a response.

We agree with the reviewer. Text has been amended as follows:

“Fig 2A shows representative montages from a stroke animal illustrating the cortical cFL and cHL Ca2+ responses to 1s, 100Hz limb stimulation of the contralateral limbs at the pre-stroke and 8week post-stroke timepoints. The location and magnitude of the cortical responses changes drastically between timepoints, with substantial loss of supra-threshold activity within the prestroke cFL representation located anterior to the cHL map, and an apparent shift of the remapped representation into regions lateral to the cHL representation at 8-weeks post-stroke. A significant decrease in the cFL evoked Ca2+ response amplitude was observed in the stroke group at 8-weeks post-stroke relative to pre-stroke (Fig 2B). This is in agreement with past studies [19–25], and suggests that cFL targeted stroke reduces forelimb evoked activity across the cFL somatosensory cortex in anaesthetized animals even after 2 months of recovery. There was no statistical change in the average size of cFL evoked representation 8-weeks after stroke (Fig 2C), but a significant posterior shift of the supra-threshold cFL map was detected (Fig 2D). Unmasking of previously sub-threshold cFL responsive cortex in areas posterior to the original cFL map at 8-weeks post-stroke could contribute to this apparent remapping. However, the amplitude of the cFL evoked widefield Ca2+ response in this distal region at 8-weeks post-stroke remains reduced relative to pre-stroke activation (Fig 2E). Previous studies suggest strong inhibition of cFL evoked activity during the first weeks after photothrombosis [25]. Without longitudinal measurement in this study to quantify this reduced activation prior to 8-weeks poststroke, we cannot differentiate potential remapping due to unmasking of the cFL representation that enhances the cFL-evoked widefield Ca2+ response from apparent remapping that simply reflects changes in the signal-to-noise ratio used to define the functional representations. There were no group differences between stroke and sham groups in cHL evoked intensity, area, or map position (data not shown).”

A lot of the nonsignificant results are reported as "statistical trends towards..." While the term "trend" is problematic, it remains common in its use. However, assigning directionality to the trend, as if it is actively approaching a main effect, should be avoided. The results aren't moving towards or away from significance. Consider rewording the way in which these results are reported.

We have amended the text to remove directionality from our mention of statistical trends.

R squared and p values for significant results are reported in the "impaired performance on tapered beam..." and "firing rate of neurons in the peri-infarct cortex..." subsections of the results, but not the other sections. Please report the results in a consistent manner.

R-squared and p-values have been removed from the results section and are now reported in figure captions consistently.

Discussion288 Remapping is defined as "new sensory-evoked spiking". This should be the main criterion for remapping, but it is not operationalized correctly by the threshold method.

With our new criterion for determining limb maps using a statistical threshold of 5X the standard deviation of baseline fluorescence, we have edited text throughout the manuscript to better emphasize that we may not be measuring new sensory-evoked spiking with the mesoscale mapping that was done. We have edited the discussion as follows:

“Here, we used longitudinal two photon calcium imaging of awake, head-fixed mice in a mobile homecage to examine how focal photothrombotic stroke to the forelimb sensorimotor cortex alters the activity and connectivity of neurons adjacent and distal to the infarct. Consistent with previous studies using intrinsic optical signal imaging, mesoscale imaging of regional calcium responses (reflecting bulk neuronal spiking in that region) showed that targeted stroke to the cFL somatosensory area disrupts the sensory-evoked forelimb representation in the infarcted region. Consistent with previous studies, this functional representation exhibited a posterior shift 8-weeks after injury, with activation in a region lateral to the cHL representation. Notably, sensory-evoked cFL representations exhibited reduced amplitudes of activity relative to prestroke activation measured in the cFL representation and in the region lateral the cHL representation. Longitudinal two-photon calcium imaging in awake animals was used to probe single neuron and local network changes adjacent the infarct and in a distal region that corresponded to the shifted region of cFL activation. This imaging revealed a decrease in firing rate at 1-week post-stroke in the peri-infarct region that was significantly negatively correlated with the number of errors made with the stroke-affected limbs on the tapered beam task. Periinfarct cortical networks also exhibited a reduction in the number of functional connections per neuron and a sustained disruption in neural assembly structure, including a reduction in the number of assemblies and an increased recruitment of neurons into functional assemblies. Elevated correlation between assemblies within the peri-infarct region peaked 1-week after stroke and was sustained throughout recovery. Surprisingly, distal networks, even in the region associated with the shifted cFL functional map in anaesthetized preparations, were largely undisturbed.”

“Cortical plasticity after strokePlasticity within and between cortical regions contributes to partial recovery of function and is proportional to both the extent of damage, as well as the form and quantity of rehabilitative therapy post-stroke [80,81]. A critical period of highest plasticity begins shortly after the onset of stroke, is greatest during the first few weeks, and progressively diminishes over the weeks to months after stroke [19,82–86]. Functional recovery after stroke is thought to depend largely on the adaptive plasticity of surviving neurons that reinforce existing connections and/or replace the function of lost networks [25,52,87–89]. This neuronal plasticity is believed to lead to topographical shifts in somatosensory functional maps to adjacent areas of the cortex. The driver for this process has largely been ascribed to a complex cascade of intra- and extracellular signaling that ultimately leads to plastic re-organization of the microarchitecture and function of surviving peri-infarct tissue [52,80,84,88,90–92]. Likewise, structural and functional remodeling has previously been found to be dependent on the distance from the stroke core, with closer tissue undergoing greater re-organization than more distant tissue (for review, see [52]).”

“Previous research examining the region at the border between the cFL and cHL somatosensory maps has shown this region to be a primary site for functional remapping after cFL directed photothrombotic stroke, resulting in a region of cFL and cHL map functional overlap [25]. Within this overlapping area, neurons have been shown to lose limb selectivity 1-month post-stroke [25]. This is followed by the acquisition of more selective responses 2-months post-stroke and is associated with reduced regional overlap between cFL and cHL functional maps [25]. Notably, this functional plasticity at the cellular level was assessed using strong vibrotactile stimulation of the limbs in anaesthetized animals. Our findings using longitudinal imaging in awake animals show an initial reduction in firing rate at 1-week post-stroke within the peri-infarct region that was predictive of functional impairment in the tapered beam task. This transient reduction may be associated with reduced or dysfunctional thalamic connectivity [93–95] and reduced transmission of signals from hypo-excitable thalamo-cortical projections [96]. Importantly, the strong negative correlation we observed between firing rate of the neural population within the peri-infarct cortex and the number of errors on the affected side, as well as the rapid recovery of firing rate and tapered beam performance, suggests that neuronal activity within the peri-infarct region contributes to the impairment and recovery. The common timescale of neuronal and functional recovery also coincides with angiogenesis and re-establishment of vascular support for peri-infarct tissue [83,97–100].”

“Consistent with previous research using mechanical limb stimulation under anaesthesia [25], we show that at the 8-week timepoint after cFL photothrombotic stroke the cFL representation is shifted posterior from its pre-stroke location into the area lateral to the cHL map. Notably, our distal region for awake imaging was directly within this 8-week post-stroke cFL representation. Despite our prediction that this distal area would be a hotspot for plastic changes, there was no detectable alteration to the level of population correlation, functional connectivity, assembly architecture or assembly activations after stroke. Moreover, we found little change in the firing rate in either moving or resting states in this region. Contrary to our results, somatosensoryevoked activity assessed by two photon calcium imaging in anesthetized animals has demonstrated an increase in cFL responsive neurons within a region lateral to the cHL representation 1-2 months after focal cFL stroke [25]. Notably, this previous study measured sensory-evoked single cell activity using strong vibrotactile (1s 100Hz) limb stimulation under aneasthesia [25]. This frequency of limb stimulation has been shown to elicit near maximal neuronal responses within the limb-associated somatosensory cortex under anesthesia [101]. Thus, strong stimulation and anaesthesia may have unmasked non-physiological activity in neurons in the distal region that is not apparent during more naturalistic activation during awake locomotion or rest. Regional mapping defined using strong stimulation in anesthetized animals may therefore overestimate plasticity at the cellular level.”

“Our results suggest a limited spatial distance over which the peri-infarct somatosensory cortex displays significant network functional deficits during movement and rest. Our results are consistent with a spatial gradient of plasticity mediating factors that are generally enhanced with closer proximity to the infarct core [84,88,90,91]. However, our analysis outside peri-infarct cortex is limited to a single distal area caudal to the pre-stroke cFL representation. Although somatosensory maps in the present study were defined by a statistical criterion for delineating highly responsive cortical regions from those with weak responses, the distal area in this study may have been a site of activity that did not meet the statistical criterion for inclusion in the baseline map. The lack of detectable changes in population correlations, functional connectivity, assembly architecture and assembly activations in the distal region may reflect minimal pressure for plastic change as networks in regions below the threshold for regional map inclusion prior to stroke may still be functional in the distal cortex. Thus, threshold-based assessment of remapping may further overestimate the neuroplasticity underlying functional reorganization suggested by anaesthetized preparations with strong stimulation. Future studies could examine distal areas medial and anterior to the cFL somatosensory area, such as the motor and pre-motor cortex, to further define the effect of FL targeted stroke on neuroplasticity within other functionally relevant regions. Moreover, the restriction of these network changes to peri-infarct cortex could also reflect the small penumbra associated with photothrombotic stroke, and future studies could make use of stroke models with larger penumbral regions, such as the middle cerebral artery occlusion model. Larger injuries induce more sustained sensorimotor impairment, and the relationship between neuronal firing, connectivity, and neuronal assemblies could be further probed relative to recovery or sustained impairment in these models. Recent research also indicates that stroke causes distinct patterns of disruption to the network topology of excitatory and inhibitory cells [73], and that stroke can disproportionately disrupt the function of high activity compared to low activity neurons in specific neuron sub-types [61]. Mouse models with genetically labelled neuronal sub-types (including different classes of inhibitory interneurons) could be used to track the function of those populations over time in awake animals. A potential limitation of our data is the undefined effect of age and sex on cortical dynamics in this cohort of mice (with ages ranging from 3-9 months) after stroke. Aging can impair neurovascular coupling [102–107] and reduce ischemic tolerance [108–111], and greater investigation of cortical activity changes after stroke in aged animals would more effectively model stroke in humans. Future research could replicate this study with mice in middle-age and aged mice (e.g. 9 months and 18+ months of age), and with sufficient quantities of both sexes, to better examine age and sex effects on measures of cortical function.”

315 - 317 Remodelling is dependent on the distance from the stroke core, with closer tissue undergoing greater reorganization than more distant tissue. There is no evidence that the more distant tissue undergoes any reorganization at all.

We agree with the reviewer that no remodelling is apparent in our distal area. We have removed reference to our study showing remodeling in the distal area, and have amended the text as follows:

“Likewise, structural and functional remodeling has previously been found to be dependent on the distance from the stroke core, with closer tissue undergoing greater re-organization than more distant tissue (for review, see [52]).”

412-414 The authors speculate that a strong stimulation under anaesthesia may unmask connectivity in distal regions. However, the motivation for this paper is that anaesthesia is a confounding factor. It appears to me that, given the results of this study, the authors should argue that the functional connectivity observed under anaesthesia may be spurious.

The incorrect word was used here. We have corrected the paragraph of the discussion and amended it as follows:

“Consistent with previous research using mechanical limb stimulation under anaesthesia [25], we show that at the 8-week timepoint after cFL photothrombotic stroke the cFL representation is shifted posterior from its pre-stroke location into the area lateral to the cHL map. Notably, our distal region for awake imaging was directly within this 8-week post-stroke cFL representation. Despite our prediction that this distal area would be a hotspot for plastic changes, there was no detectable alteration to the level of population correlation, functional connectivity, assembly architecture or assembly activations after stroke. Moreover, we found little change in the firing rate in either moving or resting states in this region. Contrary to our results, somatosensoryevoked activity assessed by two photon calcium imaging in anesthetized animals has demonstrated an increase in cFL responsive neurons within a region lateral to the cHL representation 1-2 months after focal cFL stroke [25]. Notably, this previous study measured sensory-evoked single cell activity using strong vibrotactile (1s 100Hz) limb stimulation under aneasthesia [25]. This frequency of limb stimulation has been shown to elicit near maximal neuronal responses within the limb-associated somatosensory cortex under anesthesia [101]. Thus, strong stimulation and anaesthesia may have unmasked non-physiological activity in neurons in the distal region that is not apparent during more naturalistic activation during awake locomotion or rest. Regional mapping defined using strong stimulation in anesthetized animals may therefore overestimate plasticity at the cellular level.”

FiguresFigure 1 and 2: Scale bar missing.

Scale bars added to both figures.

Figure 2: The representative image shows a drastic reduction of the forelimb response area, contrary to the general description of the findings. It would also be beneficial to see a graph with lines connecting the pre-stroke and 8-week datapoints.

The data for Figure 2 has been re-analyzed using a new criterion of 5X the standard deviation of the baseline period for determining the threshold for limb mapping. Figure 2 and relevant manuscript and figure legend text has been amended. In agreement with the reviewers observation, there is no increase in forelimb response area, but instead a non-significant decrease in the average forelimb area.